# Unsupervised Domain Adaptation for 6-DoF Pose Estimation with Contrastive Alignment and Pseudo-Label Refinement

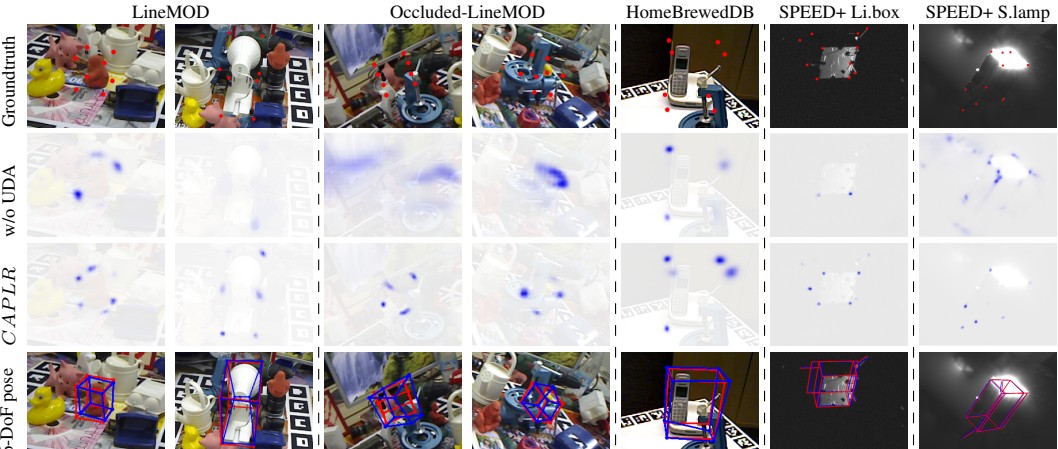

Figure 1: Qualitative results on target-domain samples, illustrating how UDA enhances heatmap predictions using the proposed $CAPLR$ framework. Row-1: object with ground-truth keypoints (red). Row-2: heatmaps from the synthetic-only model. Row-3: refined heatmaps after UDA using $CAPLR$ framework. Row-4: recovered 6-DoF pose with ground-truth and prediction overlaid.

## Abstract

Unsupervised domain adaptation (UDA) enables robust transfer of knowledge from simulated to real environments while exploiting a subset of unlabeled target data to improve real-world performance. Existing UDA methods for 6-DoF object pose estimation often rely on global feature matching, multi-stage larger frameworks, or image translation pipelines, which tend to overlook the pose-specific information embedded in feature representations. To bridge this limitation, we introduce $CAPLR$ that targets the adaptation of pose-sensitive features in localized regions, ensuring that domain alignment preserves the geometric cues essential for accurate pose estimation. $CAPLR$ achieves UDA with three key components: (1) Efficient Cross-Domain Pairing strategy leveraging intermediate features to identify pose similar image pairs across domains without supervision; (2) Contrastive Alignment to perform feature alignment at localised regions in both intermediate and task-specific representations; and (3) Consistency-Based Pseudo-Label Refinement to improve reliability by encouraging stable target predictions. Extensive experiments demonstrate that $CAPLR$ achieves state-of-the-art performance across multiple well-known 6-DoF object pose estimation benchmarks featuring diverse and challenging scenarios.

## 1 Introduction

Deep learning (DL) models trained exclusively on synthetic data often underperform on real data, primarily due to the inherent distributional discrepancies between synthetic and real domains, a phe-

nomenon commonly referred to as the *domain gap* (Denninger et al., 2020) or *sim-to-real* transfer (Pitkevich & Makarov, 2024). However, obtaining such ground-truth annotations in real-world scenarios is often impractical due to the associated costs, labor, and scalability challenges. To mitigate domain gap challenges, domain adaptation methods, particularly Unsupervised Domain Adaptation (UDA) (Farahani et al., 2021; Liu et al., 2022) were adapted to leverage any available unlabeled real-world data (i.e., target domain) along with abundant labeled synthetic data (i.e., source domain) to train DL models. UDA has shown success in classification and segmentation (Schwonberg et al., 2023); however, regression tasks such as 6-DoF object pose estimation present unique challenges due to the continuous nature of the output space, ambiguities of object symmetries, and multi-modal uncertainties (Ikeda et al., 2024). Moreover, the domain gap can include not only appearance variations, but also geometric and structural differences (Bauer et al., 2024), affecting both feature extraction and output prediction.

Recent efforts have increasingly focused on self-supervised strategies, most commonly approaches that rely on heavy training pipelines (Zhang et al., 2023) with anchor- or cluster-based partitioning entailing intricate multi-stage optimization that are sensitive to hyperparameters and often struggle with continuous and multimodal pose distributions. In contrast, image-to-image translation methods (Chen et al., 2023; Wang et al., 2023) focus on narrowing the appearance gap by rendering synthetic data more realistic; however, they cannot fully preserve the geometric or structural information essential for pose accuracy, especially when the domain gap is substantial, often resulting in noisy pseudo labels for the target domain. Multi-view consistency approaches (Tan & Dong, 2023; 2025) can enhance adaptation but rely on assumptions rarely satisfied in typical single-view scenarios. A promising but relatively underexplored direction lies in feature-level alignment, which has demonstrated strong generalization capabilities in other domains (Zhang et al., 2019a; Chen et al., 2021). Extending this concept to 6-DoF pose estimation introduces additional challenges, as indiscriminate feature alignment can disrupt task-specific spatial and geometric relationships essential for accurate pose regression, rather than enhancing them. To overcome these limitations, a practical adaptation strategy must ensure that (1) features are aligned at a fine-grained spatial scale that preserves geometric consistency, and (2) pseudo labels in the target domain remain reliable despite substantial domain-induced appearance and structural distortions.

This paper introduces $CAPLR$, a unified keypoint-based UDA framework for 6-DoF object pose estimation to address the synthetic-to-real domain gap through **C**ontrastive **A**lignment and **P**seudo-**L**abel **R**efinement, with the following key contributions: **First**, a cross-domain pairing strategy that identifies source-target pairs with consistent poses, enabling effective supervision in the absence of target annotations. **Second**, a patch-level contrastive alignment mechanism applied to both the backbone and the regression head to jointly align feature embeddings and task-specific predictions. **Third**, a consistency-based pseudo-label refinement mechanism that leverages augmented views to improve and stabilize target predictions. **Finally**, the framework's effectiveness is validated on multiple 6-DoF pose estimation benchmarks with varied domain gaps and complexities, and model sizes, complemented by an ablation study showing each component's role and contribution.

The remainder of this paper is organized as follows. Section 2 reviews the relevant literature on UDA and 6-DoF pose estimation. Section 3 introduces the proposed $CAPLR$ framework, detailing its design and implementation. Section 4 presents the experimental setup and quantitative results on benchmark datasets. Finally, Section 5 summarizes the key findings and outlines future directions.

## 2 RELATED WORKS

For DL-based 6-DoF pose estimation of known objects from an image, two widely known approaches are *Direct regression* and *Hybrid approaches*. Direct approaches aim to learn an end-to-end mapping by directly regressing the continuous pose values from an image (e.g., PoseCNN (Xiang et al., 2018)). These methods often struggle with the inherent non-linearity of the rotation space, leading to complex loss function design, lower accuracy, and limited interpretability. Hybrid methods (e.g., (Yang et al., 2024b)) predict the 2D image coordinates of predefined keypoints via a keypoint regressor; subsequently, given these 2D predictions, the known 3D coordinates of these keypoints in the object's reference frame, and the camera intrinsic parameters, Perspective-n-Point (PnP) algorithms are employed to recover the object's 6-DoF pose in the camera reference frame.

Owing to their better geometric interpretability and flexibility in incorporating spatial priors, hybrid approaches have become the dominant paradigm for pose estimation.

## 2.1 UDA FOR KEYPOINT DETECTION

The purpose of UDA is to transfer knowledge from a labeled source domain to an unlabeled target domain by addressing the distributional shift in both features and labels. UDA approaches can be broadly classified into *Reconstruction*-, *Adversarial*-, and *Discrepancy*-based methods (Farahani et al., 2021). It is beyond the scope of this paper to provide a comprehensive review of the UDA literature; however, we refer to (Farahani et al., 2021; Singhal et al., 2023) for more detailed information. UDA for keypoint detection has unique challenges, as keypoints encode precise geometric relationships, making them particularly sensitive to domain shifts (Jiang et al., 2021). The shape-consistent frameworks introduced in (Vasconcelos et al., 2021) integrate adversarial learning and self-supervision to preserve geometric structures during domain transfer. However, these approaches struggle as the domain gaps become larger, and their reliance on implicit shape priors may not be generalized to complex 3D objects. In (Jiang et al., 2021) a dual-regressor strategy inspired by the Disparity Discrepancy Theory (Zhang et al., 2019b) was introduced to stabilize adversarial training by reducing conflicts between the source and target domains. Despite their success, adversarial approaches exhibit greater training instability and sensitivity to hyperparameters. Furthermore, their indirect alignment mechanism, driven by a domain discriminator, can inadvertently align task-irrelevant or superficial features rather than exclusively task-discriminative features (Zhao et al., 2019), potentially affecting robust knowledge transfer.

Among discrepancy-based approaches, contrastive learning has emerged as an effective paradigm for classification tasks, where discrete class labels naturally define positive and negative pairs that guide discriminative representation learning (Kang et al., 2019; Xu et al., 2024). However, in regression tasks, the continuous nature of the output space makes defining such pairs challenging (Keramati et al., 2024). $CAPLR$ overcomes this limitation through a pairing strategy that establishes meaningful correspondences across domains, allowing contrastive alignment to be applied effectively for keypoint regression under domain shifts.

## 2.2 UDA FOR 6-DoF POSE ESTIMATION

Recent UDA methods for 6-DoF pose estimation focus primarily on reducing the domain gap through self-supervised learning. Typically, networks are first trained on synthetic data and then adapted using unlabeled real images. For instance, Wang et al. (2021b) proposed Self6D++, which employs self-supervised mask alignment via differentiable rendering to match predicted and ground-truth object masks. However, this approach still suffers from a noticeable discrepancy between rendered and real images, in addition to relying on an expensive rendering pipeline that limits scalability to complex scenes. Tan & Dong (2023) introduced a relative-pose geometry constraint that leverages estimated camera poses. Zhang et al. (2023) proposed MAST, a manifold-aware self-training to promote domain-invariant representations by modeling dependencies among regression targets, yet its assumption of smooth feature transitions may fail to capture local structural misalignments across domains. Other works, such as (Chen et al., 2023), attempted to bridge the gap via synthetic-to-real image translation. While this reduces appearance differences, the translation is imperfect and struggles to generalize to complex scenarios such as occlusions. More recently, Tan & Dong (2025) improved translation robustness by exploiting multi-view images with overlapping regions, but this assumption does not hold in many real-world datasets. In contrast, feature alignment has demonstrated strong performance in other UDA tasks (Zhang et al., 2019a; Chen et al., 2021), but remains largely underexplored for 6-DoF pose estimation. A key difficulty is that only task-relevant features should be aligned, as direct alignment of all representations risks amplifying superficial differences (e.g., textures, styles) while overlooking pose critical cues.

Motivated by these limitations, our work introduces a patch-based dual-level alignment strategy that jointly aligns backbone embeddings and regression head outputs, ensuring that adaptation captures both marginal and conditional distributions relevant to pose, building on prior findings that joint alignment of features and outputs improves adaptation (Zhao et al., 2022), as well as theoretical results emphasizing the importance of combining marginal and conditional alignment for regression problems (Yang et al., 2024a). Furthermore, to mitigate the unreliability of predictions in

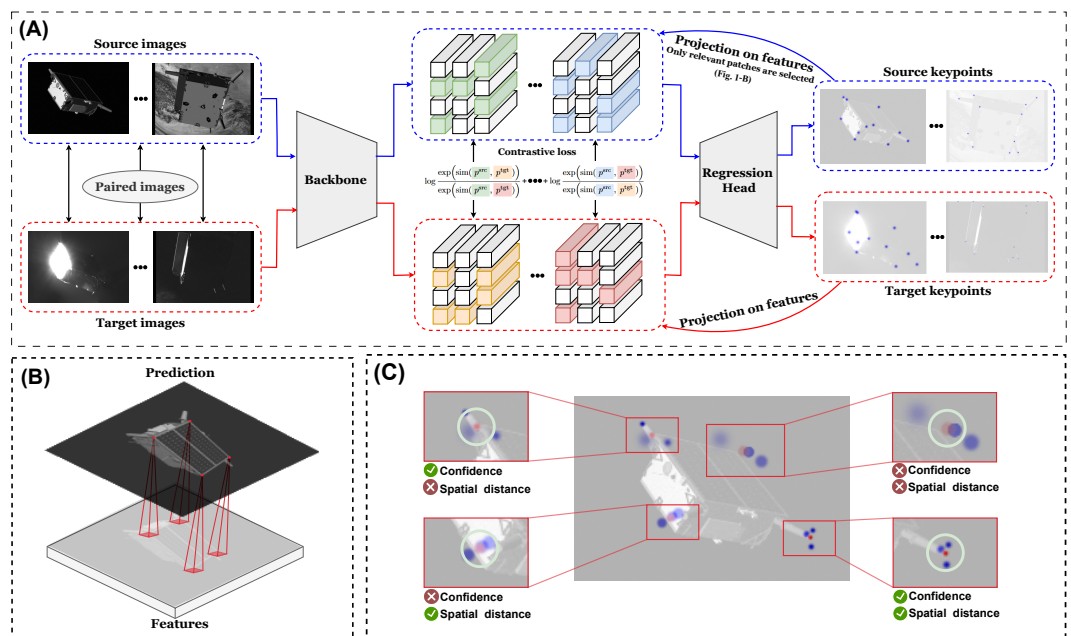

Figure 2: Overview of the proposed framework. (A) Contrastive alignment between relevant feature patches (in colors) of paired samples. (B) Keypoint projection for local patch extraction from the feature map. (C) Consistency-based filtering of pseudo-keypoints (red), incorporating confidence thresholding and spatial consistency checks (green circle).

the unlabeled target domain, we incorporate a consistency-based pseudo-label refinement that filters low-confidence or spatially inconsistent labels, thereby stabilizing supervision. Together, these components form a unified framework named $CAPLR$ that advances UDA for 6-DoF pose estimation beyond the need for image translation or computationally expensive rendering pipelines.

## 3 METHODOLOGY

### 3.1 PRELIMINARIES

Let $(x_i, y_i) \sim \mathcal{D}$ be a pair sampled from a data distribution $\mathcal{D}$ over the input space $\mathcal{X}$ and the output space $\mathcal{Y}$, where $x_i \in \mathbb{R}^{H \times W \times 3}$ is the input image and $y_i = \{(u_k, v_k)\}_{k=1}^K$ is a set of annotated 2D keypoints $K$. The objective is to train a keypoint regressor $f_\theta : \mathcal{X} \to \mathcal{Y}$ that, given an input image, predicts 2D keypoint locations by minimizing the expected prediction error, where $\mathcal{L}$ denotes a regression loss function measuring the discrepancy between the predicted and ground truth keypoints.

$$\mathcal{L}_{\text{task}}(\theta) = \mathbb{E}_{(x,y) \sim \mathcal{D}} \, \mathcal{L}(f_\theta(x), y). \tag{1}$$

Given the predicted 2D keypoints $\{(\tilde{u}_k, \tilde{v}_k)\}_{k=1}^K$, the known 3D object keypoints $\{X_k\}_{k=1}^K$, and camera intrinsics $\phi$, the optimal 6D object pose $(R^*, t^*)$ is estimated by solving a PnP problem:

$$(R^*, t^*) = \arg \min_{R,t} \sum_{k=1}^K \|\phi(RX_k + t) - (\tilde{u}_k, \tilde{v}_k)\|^2. \tag{2}$$

In the UDA approach, a labeled source domain $\hat{\mathcal{P}} = \{(x_i^s, y_i^s)\}_{i=1}^n$ is given, sampled from distribution $\mathcal{P}(x^s, y^s)$ along with an unlabeled target domain $\hat{\mathcal{Q}} = \{x_i^t\}_{i=1}^m$ sampled from the marginal $\mathcal{Q}(x^t)$ of the joint target distribution $\mathcal{Q}(x^t, y^t)$ where the labels $y^t$ are not available during training. The objective is to learn a model $f_\theta$ that minimizes the expected prediction error in the target domain:

$$\text{err}_{\mathcal{Q}} = \mathbb{E}_{(x,y) \sim \mathcal{Q}} \left[ \mathcal{L}(f_\theta(x), y) \right]. \tag{3}$$

However, this task is challenging due to the domain gap, which arises from discrepancies in both input and output distributions between the source and target domains (Yang et al., 2024a):

$$\mathcal{P}(x^s) \neq \mathcal{Q}(x^t)$$
$$\mathcal{P}(y^s \mid x^s) \neq \mathcal{Q}(y^t \mid x^t). \tag{4}$$

This mismatch hinders the generalization capability of models trained on the source domain when deployed in target domain.

## 3.2 $CAPLR$ Framework

To address the gap described in eq. (4), a domain adaptation strategy would aim to align both the input and conditional output distributions (Yang et al., 2024a) as follows:

$$\mathcal{P}(x^s) \equiv \mathcal{Q}(x^t)$$
$$\mathcal{P}(y^s \mid x^s) \equiv \mathcal{Q}(y^t \mid x^t). \tag{5}$$

However, aligning the input distributions $\mathcal{P}(x^s)$ and $\mathcal{Q}(x^t)$ is nontrivial due to domain shifts in illumination, texture, and rendering fidelity between source and target images. Similarly, aligning the conditional distributions $\mathcal{P}(y^s \mid x^s)$ and $\mathcal{Q}(y^t \mid x^t)$ is particularly difficult in regression due to the continuous nature of the output space. To overcome this, we reformulate the objective in eq. (5) within the feature space, where task-relevant representations are more amenable to alignment. Specifically, eq. (6) defines alignment over backbone features $g(x)$ and regression head outputs $h(g(x))$, which serve as proxies for the input and conditional output distributions, respectively.

$$\mathcal{P}_g\big(g(x^s)\big) \equiv \mathcal{Q}_g\big(g(x^t)\big)$$
$$\mathcal{P}_h\big(h(g(x^s))\big) \equiv \mathcal{Q}_h\big(h(g(x^t))\big), \tag{6}$$

where $\mathcal{P}_g$ and $\mathcal{P}_h$ refer to source distributions, and $\mathcal{Q}_g$ and $\mathcal{Q}_h$ to target distributions; subscripts $g$ and $h$ indicate backbone and head features, respectively. Building on this reformulation, the proposed $CAPLR$ framework instantiates the alignment process through three complementary stages: cross-domain pairing, local patch contrastive alignment, and consistency-based pseudo-label refinement. Together, these stages constitute a practical instantiation of the objective in eq. (6). The details of each stage are presented in section 3.2.1–3.2.3.

### 3.2.1 Cross-Domain Pairing

Effective feature alignment requires establishing source–target pairings with similar poses. In the UDA setting, this is particularly challenging due to the absence of target labels and the limited reliability of initial pose estimates. To address this, a two-step Cross-Domain Pairing (CDP) strategy is introduced, comprising (i) feature-based candidate selection and (ii) pose-based refinement.

***A. Feature-Based Initial Candidate Selection:*** Pose predictions obtained from a model trained exclusively on the source domain often lead to suboptimal pairings when used directly, due to errors caused by domain shift. To mitigate this limitation, the first step of CDP relies on feature embeddings rather than raw pose estimates. Although accurate keypoint localization in the target domain is unreliable, heatmap activations still provide informative cues about the approximate object region (see fig. 1-row2). These activations are therefore used to extract local features by treating predicted keypoints as anchors for cropping patches from the backbone feature map $F \in \mathbb{R}^{C \times H \times W}$.

Formally, for each image $I$, the model outputs $K$ keypoints $\{k_j = (x_j, y_j)\}_{j=1}^{K}$, each with a confidence score $c_j \in [0, 1]$ derived from the predicted heatmap. Around each keypoint $k_j$, a local patch $p_j \in \mathbb{R}^{C \times h' \times w'}$ is extracted from $F$ as shown in fig. 2-B. These patches are aggregated into a confidence-weighted image-level embedding:

$$E = \frac{\sum_{j=1}^{K} c_j \, \text{vec}(p_j)}{\sum_{j=1}^{K} c_j + \epsilon}, \tag{7}$$

where $\text{vec}(\cdot)$ denotes vectorization of the patch. Dividing by the sum of confidences normalizes the embedding ensuring that images with varying numbers of confident keypoints are represented on a similar scale. Cosine similarity is then used to cluster the resulting embeddings, forming an initial pool of structurally similar source–target image pairs.

***B. Refinement via Pose Prediction:*** The second stage refines the candidate set obtained from the previous step by applying pose-based filtering. While direct pairwise comparison of predicted poses across all source–target images is computationally expensive and error-prone due to noisy target predictions, restricting the operation to the reduced set from feature-based selection makes it feasible and more reliable. This refinement mitigates the risk of retaining structurally similar pairs that differ in pose, which would otherwise degrade feature alignment. For each candidate pair $(I_s, I_t)$ where $I_s$ and $I_t$ denote a source and target image respectively, the similarity between predicted poses $(t_s, q_s)$ and $(t_t, q_t)$ is evaluated using a combined pose error ($E_{\text{pose}}$) defined as:

$$\left.\begin{aligned}
E_T &= \|t_t - t_s\| \\
E_T^{\text{norm}} &= E_T / \|t_s\| \\
E_R &= 2\arccos(|\langle q_t, q_s \rangle|) \\
E_{\text{pose}} &= E_T^{\text{norm}} + \lambda_{\text{rot}} \cdot E_R
\end{aligned}\right\} \tag{8}$$

where $E_T$ denotes the translation error, $E_T^{\text{norm}}$ is the normalized translation error, which accounts for scale differences, and $E_R$ is the angular distance between quaternions. $\lambda_{\text{rot}}$ is a weighting factor balancing the two. The source image with the lowest $E_{\text{pose}}$ is selected as the optimal match for each target image, ensuring that only pairs with closely aligned pose are retained for the subsequent stage.

### 3.2.2 Local Patch Contrastive Alignment

Keypoint predictions in the target domain are often imprecise due to domain shift, producing dispersed heatmaps, as shown in the second row of fig. 1. Nevertheless, as established in section 3.2.1, these heatmaps frequently localize near correct object regions, providing a meaningful signal for patch-level alignment. Local Patch Contrastive Alignment (LPCA) leverages this property by aligning local feature representations across source–target pairs, in contrast to global feature alignment that may capture irrelevant background context (fig. 2-A). Specifically, feature patches are extracted around predicted keypoints (fig. 2-B) from both intermediate backbone features and pre-output regression head features. Backbone-level patches preserve spatial structure, while head-level patches encode task-specific semantics related to keypoint prediction. For a source–target pair $(I_s, I_t)$, keypoints are predicted as $\{k_j^s\}_{j=1}^K = f(I_s)$ and $\{k_j^t\}_{j=1}^K = f(I_t)$ using a network trained on synthetic data. Around each $k_j$, patches $p_j^s$ and $p_j^t$ are cropped from the feature maps. These localized descriptors are then aligned via the InfoNCE loss (Oord et al., 2018):

$$L_{\text{infoNCE}} = -\frac{1}{B \cdot K} \sum_{b=1}^{B} \sum_{k=1}^{K} \log \frac{\exp\left(\text{sim}(p_{b,k}^{\text{src}}, p_{b,k}^{\text{tgt}})/\tau\right)}{\sum_{b'=1}^{B} \exp\left(\text{sim}(p_{b,k}^{\text{src}}, p_{b',k}^{\text{tgt}})/\tau\right)}, \tag{9}$$

where $\text{sim}(\cdot)$ denotes cosine similarity, $\tau$ is a temperature hyperparameter, and $B$ is the batch size. The final alignment loss ($L_{\text{align}}$) combines the task objective with contrastive losses at both levels:

$$L_{\text{align}} = L_{\text{task}} + \lambda_{\text{backbone}} L_{\text{infoNCE}}^{\text{backbone}} + \lambda_{\text{head}} L_{\text{infoNCE}}^{\text{head}}, \tag{10}$$

where $\lambda_{\text{backbone}}$ and $\lambda_{\text{head}}$ balance the alignment and the keypoint regression objective $L_{\text{task}}$ (eq. (1)). These weights are chosen to ensure that the contrastive terms have similar magnitude to the task loss at initialization, with greater emphasis placed on backbone-level alignment due to its broader effect on domain invariance.

### 3.2.3 Consistency-Based Pseudo Label Refinement

To improve robustness on the unlabeled target domain, a Consistency-Based Pseudo Label Refinement (CBPR) strategy is applied after the LPCA stage, leveraging model confidence to identify reliable keypoint predictions and enforce consistency across multiple augmented variants of the same image. Given a target image $I_t$ a set of $M$ augmented variants $\{I_t^{(m)}\}_{m=1}^M$ is generated using the same augmentations applied to source-domain images during training. Each variant is passed through the current model $f$ producing keypoint predictions $K^{(m)} \in \mathbb{R}^{K \times 2}$ with associated confidence scores $c^{(m)} \in \mathbb{R}^K$:

$$K^{(m)}, c^{(m)} = f\left(I_t^{(m)}\right), \quad m \in \{1, \ldots, M\}. \tag{11}$$

To construct pseudo-labels, predictions across these variants are aggregated via confidence-weighted averaging. For each keypoint $k_k$, the pseudo-label is computed as follows:

$$k_{\text{pseudo}} = \frac{\sum_{m=1}^{M} c_k^{(m)} k_k^{(m)}}{\sum_{m=1}^{M} c_k^{(m)} + \epsilon}, \tag{12}$$

where $\epsilon$ ensures numerical stability. This aggregation prioritizes high-confidence predictions while down-weighting uncertain ones. Two filtering criteria are applied to improve reliability (fig. 2-C):

A. **_Confidence Thresholding:_** A pseudo-keypoint is retained only if at least one variant yields a confidence score above a threshold $c_{\text{thresh}}$:

$$\exists m \in \{1, \ldots, M\}: \quad c_k^{(m)} > c_{\text{thresh}}. \tag{13}$$

B. **_Spatial Consistency Filtering:_** Retained keypoints must also exhibit low spatial variance across variants. Specifically, the maximum distance between any prediction and the pseudo-label must remain below $d_{\text{thresh}}$:

$$\max_{m \in \{1, \ldots, M\}} \|k_k^{(m)} - k_{\text{pseudo}}\|_2 < d_{\text{thresh}}. \tag{14}$$

Only keypoints that satisfy both criteria are used as pseudo-labels and further used in a consistency loss to regularize the model. For each training image $b$ in batch $B$, with $K_b$ valid keypoints, the loss encourages predictions across all augmented variants to remain close to the pseudo-label:

$$L_{\text{pseudo}} = \frac{1}{B} \sum_{b=1}^{B} \frac{1}{K_b} \sum_{k \in K_b} \sum_{m=1}^{M} \|k_{b,k}^{(m)} - k_{b,k}^{\text{pseudo}}\|_2^2. \tag{15}$$

This loss reinforces prediction stability under diverse augmentations, effectively regularizing the model and enhancing robustness to noisy keypoints in the unlabeled target domain.

## 4 EXPERIMENTS AND RESULTS

### 4.1 EXPERIMENTAL SETUP

**_A. Datasets:_** $CAPLR$ is evaluated on different benchmarks for 6-DoF object pose estimation. LineMOD dataset (Hinterstoisser et al., 2012) contains individual sequences of 13 household objects in cluttered scenes with challenging lighting with 1.2k real images per sequence. Notably, two of the objects (Eggbox and Glue) exhibit symmetry. Following prior works, we use 15% of images for domain adaptation (without annotations) and the rest for testing. Occluded-LineMOD (Brachmann et al., 2014), a subset of LineMOD with 8 objects under severe occlusions, we use the BOP split (Hodaň et al., 2019) to sample data for testing and use the rest for adaptation. HomebrewedDB (Kaskman et al., 2019) provides new set of images of three LineMOD objects (bvise, driller, phone). This dataset differs from LineMOD in scene layouts, background appearance, camera intrinsics, and slightly varying CAD model sizes, making it a suitable benchmark to evaluate the generalization ability of $CAPLR$ to unseen variations. We use the second sequence for testing. These three real-world datasets serve as target domains, all sharing the same synthetic source domain from BOP challenge (Hodaň et al., 2020), which contains 50k PBR-generated images. For a more complex scenario, we use SPEED+ (Park et al., 2022) designed for spacecraft pose estimation, with 59,960 synthetic images (80/20 train/validation split) and two real subsets captured under varied lighting: Lightbox (6,740 image) and Sunlamp (2,791 image). For fair comparison with SOTA, all test images are used for adaptation discarding their labels. To mitigate this overlap, we also evaluate on the recent SHIRT dataset (Park & D'Amico, 2023), which provides two real sequences (roe1 and roe2) captured under lighting conditions similar to Lightbox. SHIRT serves as an external real-world validation set, enabling us to assess whether the adaptation learned on SPEED+ transfers beyond its own test imagery, and to further evaluate generalization to unseen real images.

**_B. Metrics:_** For BOP datasets, we use the Average Distance of Model Points (ADD) (Hinterstoisser et al., 2011) for asymmetric objects, which measures the average distance between 3D object points transformed by the estimated pose and those transformed by the ground truth. For symmetric objects, we use ADD-S (Xiang et al., 2018), which computes the average distance to the closest model points. Combining both, we report ADD(-S) the Average Recall (%) of poses with error below 10% of the object's diameter on all three datasets. Following (Park & D'Amico, 2024), evaluation on SPEED+ uses rotation error ($E_R$), translation and normalized translation errors ($E_T$, $E_T^{norm}$), and overall pose error ($E_{pose}$).

**_C. Network Architecture:_** Our network follows the design of Park & D'Amico (2024), consisting of an EfficientNet backbone, a Bidirectional Feature Pyramid Network (BiFPN), and a regression

Table 1: Comparison on LineMOD . The best and the second best results are in **bold** and underlined. S: synthetic data; $R$: annotated real data; $R^-$: unannotated real data. Symmetric objects are in *italic*.

| Methods | Data | Ape | Bvise. | Cam | Can | Cat | Drill | Duck | *Eggbox* | *Glue* | Holep. | Iron | Lamp | Phone | **Mean ↑** |
|---|---|---|---|---|---|---|---|---|---|---|---|---|---|---|---|
| GDR-Net (Wang et al., 2021b) | S | 50.9 | 99.4 | 89.2 | 97.2 | 79.9 | 98.7 | 24.6 | 81.1 | 81.2 | 41.9 | 98.8 | 98.9 | 64.3 | 77.4 |
| DPODv2 (Shugurov et al., 2021) | | 62.1 | 88.3 | 92.5 | 96.6 | 86.1 | 90.1 | 54.8 | 98.6 | 95.4 | 27.0 | 98.2 | 91.0 | 74.3 | 81.2 |
| MAR (Zhang et al., 2023) | | 68.6 | 97.4 | 79.4 | 98.3 | 87.1 | 94.2 | 61.3 | 82.0 | 87.1 | 56.7 | 94.3 | 92.3 | 68.8 | 82.1 |
| GDR-Net (Wang et al., 2021b) | $R$ | 85.0 | 99.8 | 96.5 | 99.3 | 93.0 | **100.0** | 65.3 | 99.9 | 98.1 | 73.4 | 86.9 | **99.6** | 86.3 | 91.0 |
| DPODv2 (Shugurov et al., 2021) | | 80.0 | 99.7 | **99.2** | 99.6 | 95.1 | 98.9 | 79.5 | 99.6 | **99.8** | 72.3 | 99.4 | 96.3 | **96.8** | 93.5 |
| Self6D++ (Wang et al., 2021a) | | 76.0 | 91.6 | 97.1 | 99.8 | 85.6 | 98.8 | 56.5 | 91.0 | 92.2 | 35.4 | 99.5 | 97.4 | 91.8 | 85.6 |
| MAST (Zhang et al., 2023) | | 73.5 | 97.2 | 80.8 | 98.6 | 89.1 | 93.9 | 66.9 | 95.3 | 95.4 | 69.8 | 95.5 | 98.9 | 93.4 | 93.2 |
| SMOC-Net (Tan & Dong, 2023) | | 85.6 | 96.7 | 97.2 | **99.9** | 95.0 | **100.0** | 76.0 | 98.3 | 99.2 | 45.6 | **99.9** | 98.9 | 94.0 | 91.3 |
| Tex-Pose (Chen et al., 2023) | $S+R^-$ | 80.9 | 99.0 | 94.8 | 99.7 | 92.6 | 97.4 | 83.4 | 94.9 | 93.4 | 79.3 | 99.8 | 98.3 | 78.9 | 91.7 |
| ONDA-Pose (Tan & Dong, 2025) | | 83.0 | **99.9** | 98.0 | 99.5 | 96.4 | 99.4 | 76.5 | 96.3 | 87.1 | 83.4 | 99.6 | 98.9 | 93.4 | 93.2 |
| $CAPLR_s(OURS)$ | | 82.1 | 95.2 | 93.2 | 97.4 | 98.8 | 97.0 | 89.5 | **100.0** | 96.9 | 95.1 | 98.8 | 78.0 | 93.7 | 93.5 |
| $CAPLR_m(OURS)$ | | **90.7** | 99.5 | 93.2 | 97.9 | **99.3** | 97.7 | **98.8** | **100.0** | 99.5 | 97.0 | **99.9** | 94.1 | 96.6 | **97.2** |

Table 2: Comparison on the Occluded-LineMOD. The top two results are in **bold** and underlined.

| Method | Data | Ape | Can | Cat | Drill | Duck | *Eggbox* | *Glue* | Holep. | **Mean ↑** |
|---|---|---|---|---|---|---|---|---|---|---|
| CosyPose (Labbé et al., 2020) | | 44.0 | 69.9 | 42.1 | 67.5 | 47.8 | 24.4 | 60.0 | 17.5 | 46.7 |
| GDR-Net (Wang et al., 2021b) | S | 44.0 | 83.9 | 49.1 | 88.5 | 15.0 | 33.9 | 75.0 | 34.9 | 52.9 |
| MAR (Zhang et al., 2023) | | 44.9 | 78.4 | 40.3 | 73.5 | 47.9 | 26.9 | 72.1 | 58.0 | 55.3 |
| Self6D++ (Wang et al., 2021a) | | 57.7 | **95.0** | 52.6 | 90.5 | 26.7 | 45.0 | 87.1 | 23.5 | 59.8 |
| MAST (Zhang et al., 2023) | | 47.6 | 82.9 | 45.4 | 75.0 | 53.7 | 48.2 | 75.3 | 63.0 | 61.4 |
| SMOC-Net (Tan & Dong, 2023) | | 60.0 | 94.5 | 59.1 | 93.0 | 37.2 | 48.3 | **89.3** | 25.0 | 63.3 |
| Tex-Pose (Chen et al., 2023) | $S+R^-$ | 60.5 | 93.4 | 56.1 | 92.5 | 55.5 | 46.0 | 82.8 | 46.5 | 66.7 |
| ONDA-Pose (Tan & Dong, 2025) | | 57.7 | 92.0 | 56.7 | 94.0 | 56.1 | 50.6 | 77.9 | 69.5 | 69.3 |
| $CAPLR_s(OURS)$ | | 62.0 | 91.3 | 61.2 | 86.1 | 69.4 | 95.6 | 75.6 | **93.6** | 79.3 |
| $CAPLR_m(OURS)$ | | **67.0** | 93.9 | **71.3** | **95.0** | **83.3** | **98.0** | 85.8 | 92.4 | **85.8** |

head with deconvolution layers that produce $K$ heatmaps, each representing one of the $K$ designated keypoints. The final 6D pose is recovered from the predicted 2D keypoints using PnP-RANSAC, which requires at least four keypoints. For the BOP datasets, all eight keypoints are used, while for the more challenging SPEED+ dataset, only the top five keypoints (ranked by prediction confidence) are employed, yielding better performance. The robustness of $CAPLR$ is evaluated across model scales using two variants: $CAPLR_s$, based on EfficientNet-B0 with ~3.9M parameters, and $CAPLR_m$, based on EfficientNet-B5 with ~35M parameters.

### 4.2 COMPARATIVE EVALAUTION

***A. LineMOD:*** The results are compared against a range of methods employing different levels of supervision, including models trained solely on synthetic data (GDR-Net (Wang et al., 2021b), DPODv2 (Shugurov et al., 2021) and MAR (Zhang et al., 2023)), models trained on annotated real data (GDR-Net and DPODv2), and models trained on synthetic data combined with unlabeled real images (Self6D++ Wang et al. (2021a), MAST (Zhang et al., 2023), SMOC-Net (Tan & Dong, 2023), Tex-pose (Chen et al., 2023) and ONDA-pose (Tan & Dong, 2025)). Detailed results are reported in Table 1. Notably, $CAPLR_s$ alone outperforms all methods and even reaches performance comparable to DPODv2, which leverages annotated real images during training. $CAPLR_m$ achieves even stronger results, demonstrating both the effectiveness of $CAPLR$ for domain adaptation in 6D object pose estimation and its robustness to model size.

***B. Occluded-LineMOD:*** Results are compared against methods trained solely on synthetic images (CosyPose (Labbé et al., 2020), GDR-Net and MAR) as well as those leveraging unlabeled real data (Self6D++, MAST, SMOC-Net, Tex-pose and ONDA-pose). As shown in Table 2, approaches that combine synthetic and unlabeled real images generally reduce the domain gap and outperform purely synthetic trained models. Nevertheless, $CAPLR$ surpasses all UDA baselines by a margin of at least 10% in ADD(-S) score, demonstrating the effectiveness of the framework even under the challenging conditions of severe occlusions.

***C. SPEED+:*** The results are presented in Table 3, with performance further analyzed by model size, an aspect of particular importance in space applications. Comparisons are made against the top three winners of the SPEC21[1] challenge as well as recent state-of-the-art approaches (Yolov8s (Bechini & Lavagna, 2025), UAKD (Ali Ousalah et al., 2025), SPNv2 (Park & D'Amico, 2024), PVSPE (Yang et al., 2024b) and FA-VAE (Yanfang et al., 2024)). Notably, Yolov8s, UAKD, PVSPE and FA-VAE

---

[1]https://kelvins.esa.int/pose-estimation-2021/

Table 3: Comparison of pose estimation performance across target domains in the SPEED+. Results marked with † are methods using S+R⁻ data, * correspond to top-winning methods of SPEC21.

| Method | Size (M) | Lightbox $E_T[m]$ / $E_T^{norm}[.]\downarrow$ | $E_R[°]\downarrow$ | $E_{pose}[.]\downarrow$ | Sunlamp $E_T[m]$ / $E_T^{norm}[.]\downarrow$ | $E_R[°]\downarrow$ | $E_{pose}[.]\downarrow$ | Mean $E_{pose}[.]\downarrow$ |
|---|---|---|---|---|---|---|---|---|
| *Small models* | | | | | | | | |
| Lava1302 Wang et al. (2023) *† | 8.2 | - / 0.048 | **6.66** | **0.165** | - / **0.011** | **2.73** | **0.059** | **0.112** |
| YOLOv8s Bechini & Lavagna (2025) | 11.5 | 0.758 / 0.118 | 18.0 | 0.432 | 0.518 / 0.086 | 19.8 | 0.432 | 0.432 |
| UAKD Ali Ousalah et al. (2025) | 3.8 | 0.288 / 0.049 | 11.42 | 0.248 | 0.373 / 0.063 | 17.03 | 0.360 | 0.304 |
| $CAPLR_s(OURS)$ † | 3.9 | **0.173 / 0.029** | 7.80 | **0.165** | 0.215 / 0.035 | 11.17 | 0.230 | 0.198 |
| *Large/Medium models* | | | | | | | | |
| TangoUnchained Park et al. (2023) *† | 50 | - / 0.018 | **3.19** | **0.073** | - / 0.015 | 4.30 | 0.090 | 0.082 |
| VPU Pérez-Villar et al. (2022) *† | 190 | - / 0.022 | 4.58 | 0.101 | - / **0.012** | **2.83** | **0.061** | **0.081** |
| SPNv2 Park & D'Amico (2024) † | 53 | 0.150 / 0.024 | 5.58 | 0.122 | 0.161 / 0.030 | 9.79 | 0.197 | 0.160 |
| PVSPE Yang et al. (2024b) | 70 | - / **0.017** | 4.81 | 0.101 | - / 0.022 | 8.94 | 0.178 | 0.140 |
| FA-VAE Yanfang et al. (2024) | 42 | - / 0.027 | 4.94 | 0.114 | - / 0.028 | 5.19 | 0.118 | 0.116 |
| $CAPLR_m(OURS)$ † | 35 | 0.103 / 0.018 | 3.54 | 0.080 | 0.126 / 0.022 | 5.77 | 0.122 | 0.101 |

Table 4: Generalization Results of $CAPLR$ Across Domains, Adapted on LineMOD and SPEED+, Evaluated on HomebrewedDB and SHIRT datasets

| HomeBrewedDB | | | | | SHIRT | | | | | |
|---|---|---|---|---|---|---|---|---|---|---|
| Method | Bvise | Drill | Phone | Mean ↑ | Method | Data | $E_T[m]\downarrow$ roe1 | $E_R[°]\downarrow$ | $E_T[m]\downarrow$ roe2 | $E_R[°]\downarrow$ |
| Self6D++ (Wang et al., 2021a) | 75.7 | 89.4 | 76.8 | 80.6 | Cassinis et al. (2023) | $S+A$ | **0.250**±0.06 | 14.00±11.00 | 0.380±0.06 | 9.00±7.50 |
| MAST (Zhang et al., 2023) | 93.8 | 91.5 | 81.8 | 89.0 | Park & D'Amico (2023) | $S+B$ | 0.303±0.35 | 17.17±42.25 | 0.203±0.23 | 4.67±14.90 |
| Tex-Pose (Chen et al., 2023) | 93.1 | 94.8 | 79.3 | 89.1 | | $S$ | 0.348±2.30 | 10.74±27.77 | 0.110±0.30 | 6.04±22.23 |
| ONDA-Pose (Tan & Dong, 2025) | **98.2** | **97.1** | 92.1 | **95.8** | $CAPLR_m(OURS)$ | $S+R^-$ | 0.266±0.78 | 10.65±28.02 | **0.087**±0.17 | 4.49±17.79 |
| $CAPLR_m(OURS)$ | 93.0 | 91.5 | **94.1** | 92.9 | | $S+R$ | 0.269±1.41 | **10.31**±28.02 | 0.097±0.42 | **3.74**±14.27 |

are trained exclusively on synthetic images, whereas SPNv2 and the top SPEC21 winners leverage both synthetic and unlabeled real images, similar to our setup. For smaller models, $CAPLR_s$ proves highly competitive, achieving the best performance in the Lightbox domain and the second-best in the Sunlamp domain, despite having roughly half the size of the leading model (Lava1302). For medium and larger models, $CAPLR_m$ continues to perform strongly, securing the second-best result in Lightbox, outperforming models up to five times larger, such as VPU, and the fourth-best in Sunlamp. When averaging performance across both domains, $CAPLR_m$ ranks third overall after VPU and TangoUnchained models highlighting that $CAPLR$ achieves consistently strong results across varied real-world space conditions.

***D. Generalization to unseen domains:*** Direct feature alignment may raise concerns regarding generalization to unseen domains. To evaluate this, we assess two setups: (i) $CAPLR_m$ models adapted using only 15% of real LineMOD images, tested on HomebrewedDB; and (ii) models adapted on the Lightbox domain of SPEED+, tested on the SHIRT dataset. Results are summarized in Table 4. On HomebrewedDB (left), $CAPLR_m$ achieves the second-best overall performance compared with Self6D++, MAST, Tex-Pose, and ONDA-Pose, demonstrating strong robustness to changes in scene layout, background, and camera intrinsics. On SHIRT (right), we compare against (Cassinis et al., 2023; Park & D'Amico, 2023), where $CAPLR_m$ attains superior performance. We also report baselines trained only on synthetic data and on synthetic+real SPEED+ images as lower and upper bounds, the performance gap notably narrows, indicating that $CAPLR_m$ effectively transfers beyond its adaptation domain.

## 4.3 ABLATION STUDIES

To assess the design choices in our framework and quantify the contribution of its components, we conduct a comprehensive ablation study, as detailed below.

***A. Effectiveness of Two-Stage Pairing Strategy:*** The cross-domain pairing approach consists of two stages: initial selection of candidate based on features followed by refinement based on poses. To assess the effectiveness of this two-step strategy, comparisons are made against single-stage alternatives and ground-truth-based pairings as an upper bound. To measure pair quality, we define a compatibility metric based on pose error thresholds, as shown below:

$$\text{Compatible}(I_s, I_t) = \begin{cases} 1, & \text{if } d_{\text{rot}}(q_s, q_t) \leq \text{thresh}_{\text{rot}} \text{ and } d_{\text{tran}}(t_s, t_t) \leq \text{thresh}_{\text{tran}} \\ 0, & \text{otherwise} \end{cases} \quad (16)$$

Based on the above eq. (16), we calculate the number of compatible pairs from the total pairs as the accuracy reported in Table 5 that shows that the two-stage strategy significantly outperforms the

Table 5: CDP performance evaluated with the proposed compatibility metric across varying distance and orientation thresholds, comparing Ground-Truth pairs, Single-Stage pairs (features-only or predictions-only), and Two-Stage pairs.

| Method | Lightbox - Accuracy(%)↑ | | | | Sunlamp - Accuracy(%)↑ | | | |
| --- | --- | --- | --- | --- | --- | --- | --- | --- |
| | $[0.5\,m, 10°]$ | $[1\,m, 10°]$ | $[0.5\,m, 15°]$ | $[1\,m, 15°]$ | $[0.5\,m, 10°]$ | $[1\,m, 10°]$ | $[0.5\,m, 15°]$ | $[1\,m, 15°]$ |
| Ground-Truth pairs | 47.06 | 59.23 | 79.79 | 88.50 | 48.15 | 55.18 | 80.94 | 89.29 |
| Single-Stage (Features) | 7.39 | 16.71 | 16.22 | 35.16 | 5.98 | 12.76 | 13.62 | 28.84 |
| Single-Stage (Predictions) | 36.90 | 44.79 | **64.66** | 75.95 | 32.32 | 38.45 | **56.5** | 66.46 |
| Two-stage | **37.83** | **56.88** | 59.63 | **83.26** | **32.78** | **46.76** | 52.38 | **72.62** |

Table 6: *(Left)* Comparison of feature-level alignment: Backbone-only (SL1), Head-only (SL2), and Dual-Level (DL). *(Right)* Impact of limited target-domain data (10%, 20%, 100%) on performance.

| Configuration | Lightbox-$E_{pose}[-]\downarrow$ | Sunlamp-$E_{pose}[-]\downarrow$ |
| --- | --- | --- |
| $CAPLR_s(SL1)$ | 0.166 | 0.384 |
| $CAPLR_s(SL2)$ | 0.177 | 0.260 |
| $CAPLR_s(DL)$ | 0.165 | 0.230 |

| Configuration | Lightbox-$E_{pose}[-]\downarrow$ | Sunlamp-$E_{pose}[-]\downarrow$ |
| --- | --- | --- |
| $CAPLR_s^{+10\%}$ | 0.179 | 0.291 |
| $CAPLR_s^{+20\%}$ | 0.179 | 0.259 |
| $CAPLR_s^{+100\%}$ | 0.165 | 0.230 |

Table 7: Impact of the Consistency-Based Pseudo Label Refinement Stage. $CA_S$ indicates results obtained using only the Local Patch Contrastive Alignment stage.

| Configuration | LineMOD $ADD(-S)[\%]\uparrow$ | Occluded-LineMOD $ADD(-S)[\%]\uparrow$ | Lightbox $E_{pose}[-]\downarrow$ | Sunlamp $E_{pose}[-]\downarrow$ |
| --- | --- | --- | --- | --- |
| baseline (Syn. only) | 84.4 | 70.3 | 0.193 | 0.395 |
| + $CA_s$ | 93.3 | 78.7 | 0.177 | 0.256 |
| + $CAPLR_s$ | 93.5 | 79.3 | 0.165 | 0.230 |

single-stage methods and approaches the performance of ground truth-based pairings, confirming the effectiveness of the two-stage pairing strategy.

***B. Effectiveness of Dual-Level Alignment:*** The proposed dual alignment strategy simultaneously aligns backbone and head features across source and target domains, bridging the gap at multiple levels of representation. To evaluate the effectiveness of this approach, we compare configurations that perform alignment exclusively on either backbone (SL1) or head (SL2) features against the dual-level (DL) alignment strategy. As shown in Table 6 (*Left*), the dual-level alignment consistently outperforms partial variants, underscoring the complementary benefits of aligning both feature types.

***C. Generalizability with Limited Target Data:*** To assess the effectiveness of $CAPLR$ under limited target data availability, experiments were conducted using subsets of 10% and 20% of the target images, with performance evaluated on the full target test set. The results, presented in Table 6 (*Right*), demonstrate that $CAPLR$ significantly mitigates the domain gap even with substantially reduced target supervision.

***D. Effectiveness of the CBPR Stage:*** We evaluate the impact of incorporating the Consistency-Based Pseudo Label Refinement (CBPR) stage by comparing the full CAPLR framework with a version that omits CBPR (denoted as $CA$). While $CA$ alone achieves strong performance, the addition of the CBPR stage consistently improves results across multiple datasets, as shown in Table 7, confirming the effectiveness of this refinement step in enhancing keypoint prediction robustness.

## 5 CONCLUSION

This paper introduced $CAPLR$, a unified keypoint-based unsupervised domain adaptation framework for 6-DoF object pose estimation, which integrates cross-domain pairing, contrastive alignment, and pseudo-label refinement. Unlike existing self-supervised methods, $CAPLR$ addresses domain gaps through explicit feature alignment. We evaluated $CAPLR$ on diverse domains with varying complexities and across different model sizes, demonstrating state-of-the-art performance on multiple benchmarks. Despite its effectiveness, $CAPLR$ has some limitations: it relies on finding pairings between source and target images, which may be challenging when synthetic CAD models differ significantly from real ones, and its patch extraction process depends on uncertain predictions rather than being fully learnable. Future work could address these limitations by focusing on eliminating the need for explicit pairings and exploring learning-based patch extraction.

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

# 6 APPENDIX

## 6.1 ADDITIONAL ANALYSIS

### 6.1.1 IMPACT OF SYNTHETIC DOMAIN PRETRAINING

We evaluate the effect of synthetic pretraining on the SPEED+ dataset by comparing two settings prior to adaptation with $CAPLR_s$ : (i) initializing the backbone (EfficientNet, as described in Section 4.1) with ImageNet-pretrained weights and adapting only the regression head, and (ii) pretraining the entire network on the full synthetic dataset before adaptation. table 8 shows that synthetic pretraining of the full network results in a significant improvement in post-adaptation performance compared to ImageNet-only pretraining, confirming that pretraining on the synthetic domain is essential.

Table 8: Comparison of performance with partial (i) and full (ii) network synthetic domain pretraining on the Lightbox and Sunlamp domains.

| Config. | Lightbox $E_{pose}[-]\downarrow$ | Sunlamp $E_{pose}[-]\downarrow$ |
|---|---|---|
| (i) | 1.65 | 2.38 |
| (ii) | 0.165 | 0.230 |

### 6.1.2 EFFECTIVENESS OF CONSISTENCY-BASED PSEUDO-LABEL REFINEMENT

The consistency-based pseudo-label Refinement (CBPR) stage constitutes the final stage of $CAPLR$ and is specifically designed to enhance the model's prediction confidence on the target domain. This process is executed over $N$ training epochs ($N = 40$ in our experiments), and for each run, we select the epoch yielding the highest number of accepted pseudo-keypoints, determined according to the criteria described in 3.2.3. In fig. 3a and 3b, the blue curve shows the maximum average number of accepted keypoints per 100 images achieved at this stage, revealing a clear upward trend. The red curve depicts the best linear fit computed across all epochs (not restricted to the selected best ones), and likewise indicates a consistent improvement across both domains. Together, these results demonstrate a steady increase in model confidence, as captured by the pseudo-label selection criteria.

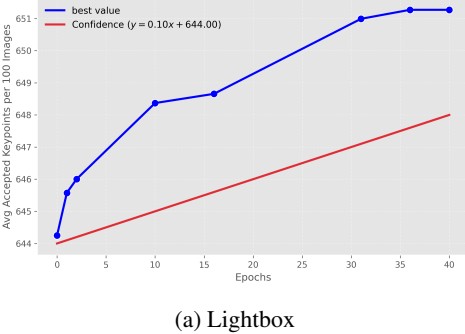

(a) Lightbox

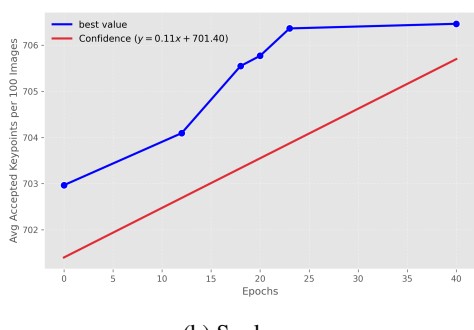

(b) Sunlamp

Figure 3: Progression of accepted pseudo-labeled keypoints across epochs (blue) following the two criterias defined in 3.2.3. The red line indicates the linear trend fitted to the data.

### 6.1.3 VALIDATION PERFORMANCE ACROSS CBPR STAGE

During the CBPR stage, training relies solely on the loss between predicted keypoints on the augmented images and the pseudo labels, omitting direct task supervision. This could raise concerns about training stability and potential performance degradation. To investigate this, we report the percentage of correct keypoints (within 5% of image width) on the SPEED+ domains during this stage, shown in Figure 4. The results exhibit only minor fluctuations, indicating that performance remains largely stable throughout CBPR.

This stability is primarily ensured by two factors. First, the augmentations used are simple and similar to those seen during training, preventing the model from encountering overly unfamiliar inputs. Second, update acceptance conditions based on prediction confidence and spatial consistenc, filter out unreliable pseudo-label updates, avoiding detrimental updates from poorly aligned keypoints. Together, these mechanisms maintain stable performance across CBPR epochs.

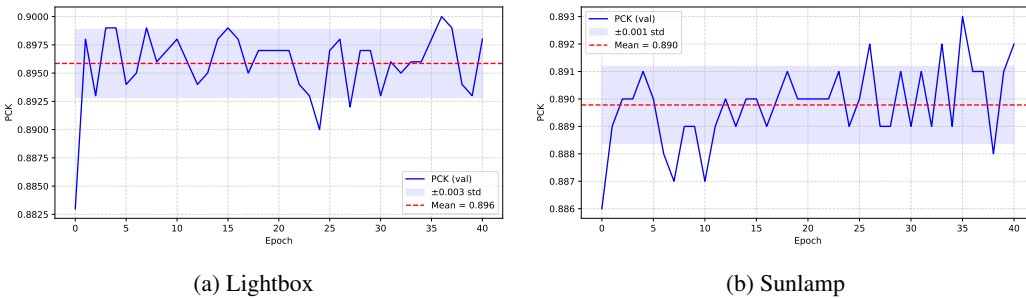

(a) Lightbox                                      (b) Sunlamp

Figure 4: Validation performance (PCK, 5% of image width) on real images during the CBPR stage. The performance remains stable across epochs.

### 6.1.4 EXTENDED RESULTS AND ABLATION ON $CAPLR$ PERFORMANCE

To better illustrate the gains attributable to $CAPLR$, we report in this section the lower and upper bounds of model performance. The lower bound corresponds to the model trained only on synthetic images, while the upper bound corresponds to a model trained on synthetic images and fine-tuned on real images for the same number of alignment epochs as in $CAPLR$. Results are reported on LineMOD (Table 9), Occluded-LineMOD (Table 10), and SPEED+ (Table 11). For the first two datasets, the real images used for fine-tuning are the same as those used for adaptation; for SPEED+, we use the first 20% of real images.

As evident from tables, $CAPLR$ achieves consistent and substantial gains over the synthetic-only baseline. Moreover, in most cases, the improvement from synthetic-only to $CAPLR$ is larger than the difference between the upper-bound (syn+real) model and $CAPLR$, highlighting the effectiveness of $CAPLR$ in closing the gap towards real-data performance.

Table 9: Results on LineMOD – Baseline (S), $CAPLR$ (S+R$^-$), and Real-Finetuned Model (S+R)

| Method | Data | Ape | bvise | cam | Can | Cat | Drill | Duck | *Eggbox* | *Glue* | Holep | iron | lamp | phone | **Mean ↑** |
|--------|------|-----|-------|-----|-----|-----|-------|------|----------|--------|-------|------|------|-------|------------|
| | S | 49.7 | 94.5 | 74.1 | 93.3 | 95.1 | 94.3 | 78.3 | 96.9 | 84.2 | 91.1 | 98.7 | 63.8 | 82.8 | 84.4 |
| $CAPLR_s$ | S+R$^-$ | 82.1 | 95.2 | 93.2 | 97.4 | 98.8 | 97.0 | 89.5 | 100.0 | 96.9 | 95.1 | 98.8 | 78.0 | 93.7 | 93.5 |
| | S+R | 95.1 | 98.9 | 95.1 | 95.8 | 93.1 | 99.7 | 96.3 | 100 | 96.1 | 98.7 | 99.8 | 86.6 | 96.8 | 96.3 |
| | S | 75.2 | 99.5 | 85.1 | 95.6 | 96.6 | 95.1 | 96.0 | 99.8 | 84.6 | 92.9 | 99.7 | 75.2 | 90.9 | 90.8 |
| $CAPLR_m$ | S+R$^-$ | 90.7 | 99.5 | 93.2 | 97.9 | 99.3 | 97.7 | 98.8 | 100.0 | 99.5 | 97.0 | 99.9 | 94.1 | 96.6 | 97.2 |
| | S+R | 98.1 | 99.9 | 97.7 | 98.4 | 100 | 99.9 | 99.9 | 100 | 98.6 | 99.7 | 100 | 96.8 | 99.9 | 99.2 |

Table 10: Results on Occluded-LineMOD – Baseline (S), $CAPLR$ (S+R$^-$), and Real-Finetuned Model (S+R)

| Method | Data | Ape | Can | Cat | Drill | Duck | *Eggbox* | *Glue* | Holep | **Mean ↑** |
|--------|------|-----|-----|-----|-------|------|----------|--------|-------|------------|
| | S | 54.6 | 80.3 | 56.1 | 79.6 | 55.8 | 91.1 | 62.9 | 80.3 | 70.3 |
| $CAPLR_s$ | S+R$^-$ | 62.0 | 91.3 | 61.2 | 86.1 | 69.4 | 95.6 | 75.6 | 93.6 | 79.3 |
| | S+R | 85.0 | 95.8 | 87.7 | 90.3 | 82.9 | 95.3 | 80.6 | 98.4 | 89.5 |
| | S | 54.6 | 84.0 | 54.5 | 92.5 | 79.5 | 97.4 | 68.6 | 86.7 | 77.2 |
| $CAPLR_m$ | S+R$^-$ | 67.0 | 93.9 | 71.3 | 95.0 | 83.3 | 98.0 | 85.8 | 92.4 | 85.8 |
| | S+R | 90.0 | 97.4 | 95.1 | 96.8 | 88.4 | 98.4 | 87.9 | 99.0 | 94.1 |

Table 11: Results on SPEED+ – Baseline (S), $CAPLR$ (S+R$^-$), and Real-Finetuned Model (S+R)

| | **DATA** | **Lightbox** | | | **Sunlamp** | | | **Mean** |
|--------|----------|--------------------------------|---------------|------------------|--------------------------------|---------------|------------------|------------------|
| Method | (M) | $E_T[m] / E_T^{norm}[.]\downarrow$ | $E_R[°]\downarrow$ | $E_{pose}[.]\downarrow$ | $E_T[m] / E_T^{norm}[.]\downarrow$ | $E_R[°]\downarrow$ | $E_{pose}[.]\downarrow$ | $E_{pose}[.]\downarrow$ |
| | S | 0.221 / 0.036 | 8.98 | 0.193 | 0.426/0.065 | 18.93 | 0.395 | 0.294 |
| $CAPLR_s$ | S+R$^-$ | 0.173 / 0.029 | 7.80 | 0.165 | 0.215 / 0.035 | 11.17 | 0.230 | 0.198 |
| | S+R | 0.127 / 0.020 | 4.74 | 0.103 | 0.131 / 0.021 | 5.38 | 0.115 | 0.109 |
| | S | 0.162 / 0.026 | 5.31 | 0.119 | 0.288 / 0.044 | 11.09 | 0.237 | 0.178 |
| $CAPLR_m$ | S+R$^-$ | 0.103 / 0.018 | 3.54 | 0.080 | 0.126 / 0.022 | 5.77 | 0.122 | 0.101 |
| | S+R | 0.097 / 0.016 | 2.99 | 0.068 | 0.106 / 0.017 | 3.69 | 0.082 | 0.075 |

## 6.2 TECHNICAL DETAILS

### 6.2.1 NETWORK ARCHITECTURE AND TRAINING

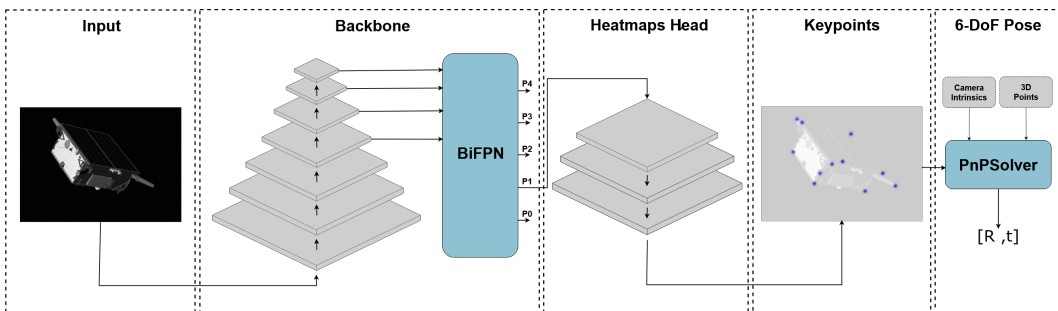

Figure 5: Architecture used in experiments, comprising an EfficientNet backbone, a BiFPN layer, and a heatmap head with three deconvolution layers.

The adopted architecture consists of an EfficientNet backbone, followed by a Bidirectional Feature Pyramid Network (BiFPN) and a regression head with three deconvolution layers that generate $K$ heatmaps, each corresponding to one of the $K$ annotated keypoints (fig. 5). The BiFPN produces features at multiple levels, each capturing different amounts of information. For the CDP stage, we use the P4 features due to their large receptive fields and robustness to stylistic variations (Park & D'Amico, 2024). For the alignment, we rely on the P1 features, as they directly drive the final predictions.

For the BOP datasets, training is performed exclusively on synthetic images without any data augmentation. Input images are padded by 16 pixels on the top and bottom (resulting in a resolution of $640 \times 512$). The corresponding output heatmaps are generated at a resolution of $320 \times 256$. Training is conducted for 5 epochs using the AdamW optimizer (Loshchilov & Hutter, 2019), with a learning rate of $10^{-4}$ and a batch size of 16.

For the SPEED+ dataset, synthetic data is also used, but augmented with the transformations proposed in (Park & D'Amico, 2024). Input images are padded by 40 pixels on the top and bottom to match the original image ratio ($1920 \times 1280$), and resized to $768 \times 512$. The heatmap resolution is set to $384 \times 256$. Training follows the same optimizer and batch size settings, but is extended to 80 epochs.

All experiments are conducted on a single NVIDIA A6000 GPU.

During the Cross-Domain Pairing (CDP) stage, the number of initial candidates is fixed at 1000, with patch sizes of $12 \times 12$ ($10 \times 10$) for backbone features and $10 \times 10$ ($10 \times 10$) for head features on SPEED+ (BOP datasets), as this configuration yields the best empirical performance. We do not follow a strict rule for choosing the patch size; rather, we observe that it seems to depend on how well the synthetic-trained model performs on the real domain, where better performance allows the use of smaller patches, and also on the object size, with smaller objects tending to favor smaller patches, The weighting parameter $\lambda_{rot}$ in eq. (8) is set to 2 to give more emphasis to orientation differences, as the patch-level alignment is inherently more sensitive to angular variations than to translational ones. In the Local Patch Contrastive Alignment (LPCA) stage, the contrastive loss usually converges within five epochs. The model is evaluated on the target domain at each epoch, and the checkpoint with the highest percentage of correct keypoints score is retained.

Finally, the Consistency-Based Pseudo-Refinement (CBPR) stage is performed exclusively on the target domain. We use the augmentation set consisting of Blur, Emboss, and FancyPCA from the Albumentations library. Training runs for 40 epochs, and the checkpoint selected for final evaluation corresponds to the one that yields the highest number of accepted keypoints under the consistency criterion defined in section 3.2.3.

### 6.2.2 LOSS FUNCTION CHOICE

***A. Heatmap Supervision with KL Divergence Loss:***

The heatmap regression model outputs $K$ heatmaps, each corresponding to one of the $K$ keypoints. To supervise the heatmap regression, the Kullback-Leibler divergence (KLD) loss is used. The KLD

loss makes the predicted heatmaps more uniformly distributed around the correct keypoint locations by reducing spatial divergence (Luo et al., 2021). An alternative approach is to extract keypoints from the predicted heatmaps (e.g., using argmax or soft-argmax) and compute the mean squared error (MSE) between the predicted and ground-truth keypoint coordinates. Two training strategies were initially explored: (a) using only the KLD loss and (b) combining KLD with the MSE loss. In the combined setting, a weighting factor $\lambda$ was introduced to balance the two losses to ensure that they operate on a similar scale. However, MSE supervision at the coordinate level may weaken the training signal by smoothing gradients or encouraging less sharply defined heatmaps, as it does not directly enforce spatial distribution alignment. In contrast, KLD loss alone promotes accurate localization and well-formed heatmaps. The results of training with synthetic data (Table 12) indicate that using KLD loss alone produces the best performance, supporting its selection as the final loss function.

Table 12: Pose Error ($E_{pose}$) Comparison of performance using KLD loss alone versus a combination of KLD on heatmaps and MSE on keypoints.

| Configuration | Lightbox - $E_{pose}[-] \downarrow$ | Sunlamp - $E_{pose}[-] \downarrow$ |
|---|---|---|
| KLD | 0.193 | 0.395 |
| KLD + $\lambda$MSE | 0.257 | 0.512 |

**B. Contrastive Alignment with InfoNCE Loss:**

To align the features between the synthetic and real domains, InfoNCE loss is used due to its strong theoretical foundation and practical effectiveness in learning contrastive representation. InfoNCE maximizes a lower bound on mutual information between positive pairs, promoting semantic consistency across domains while distinguishing them from unrelated negative samples (Oord et al., 2018). Its softmax-based formulation over large negative sets generally ensures stable gradients and efficient use of negatives, which is particularly beneficial in unsupervised domain adaptation, where explicit supervision is not available. Compared to alternative contrastive losses, such as triplet loss, InfoNCE benefits from using many negatives per anchor without the need for hard negative mining or manually tuned margins, leading to more robust and scalable training. While losses like NT-Xent (used in SimCLR) are also effective, they are essentially temperature-scaled and normalized variants of InfoNCE (Chen et al., 2020). However, InfoNCE is preferred here because of its interpretability in terms of mutual information maximization and its ease of integration into the alignment stage of our framework.

### 6.2.3 HEATMAPS GENERATION FOR TRAINING

For keypoint regression, the ground-truth 2D coordinates are converted into target heatmaps using a Gaussian rendering function. For each keypoint, a 2D Gaussian distribution centered at its annotated location is generated with the $render\_gaussian2d$ function from the DSNT library:

```
target = dsnt.render_gaussian2d(mean=keypoints, std=[1,1], size=(384,256))
```

This procedure yields one heatmap per keypoint, where the peak intensity encodes the confidence of the keypoint localization. The peak value ranges between 0 and 1, serving as a confidence score. During inference, the predicted keypoint location is obtained by identifying the pixel with maximum value in the corresponding heatmap.

### 6.2.4 KEYPOINT EXTRACTION

Predicted keypoints are extracted from the output heatmaps through the following procedure:

A. *Initial Localization:* Identification of coordinates corresponding to the maximum value in each heatmap, referred to as coarse keypoints.

B. *Subpixel Refinement:* Enhancement of coarse keypoints via second-order Taylor expansion applied on a local 3×3 patch around each coarse prediction. This refinement technique improves localization accuracy beyond the heatmap resolution, as demonstrated in Table

13. The refinement approach adapted from (Xiao et al., 2018) calculates a subpixel offset utilizing image gradients and a local quadratic approximation as detailed in algorithm 1 below.

---

**Algorithm 1** Subpixel Keypoint Refinement

---

**Require:** A set of predicted keypoints and their heatmaps
**Ensure:** Refined keypoints
1: **for all** predicted keypoint $k$ **do**
2:     Extract a $3 \times 3$ patch from the heatmap around $k$
3:     Compute gradients $d_x, d_y$ and Hessian matrix $H$ from the patch
4:     **if** $H$ is invertible **then**
5:         Compute subpixel offset :    $\Delta = H^{-1} \left[ -d_x - d_y \right]$
6:         Refine keypoint:    $k \leftarrow k + \Delta$
7:     **end if**
8: **end for**
9: **return** refined keypoints

---

### 6.2.5   6-DoF Pose Recovery

To estimate the 6-DoF pose from the predicted keypoints, the Perspective-n-Point (PnP) algorithm with RANSAC is initially employed, followed by pose refinement. The procedure is as follows:

A. ***Initial Estimation:*** The initial pose estimation is obtained using the `solvePnPRansac` function from OpenCV, which estimates the rotation and translation vectors by minimizing reprojection errors under potential outliers.

B. ***Refinement:*** To enhance accuracy, the initial rotation and translation estimates from `solvePnPRansac` are further refined using the Levenberg-Marquardt optimization algorithm via the `solvePnPRefineLM` function, improving the precision of pose estimation, as validated in Table 13.

Table 13: Advantage of keypoint and pose refinement on pose error, comparing the performance of $CAPLR_s$ with and without refinement on the SPEED+ real domains.

| Configuration | Lightbox $E_{pose}[-] \downarrow$ | Sunlamp $E_{pose}[-] \downarrow$ |
|---|---|---|
| w/o refine | 0.179 | 0.248 |
| w/ keypoints refine | 0.176 | 0.247 |
| w/ pose refine | 0.166 | 0.232 |
| w/ keypoints & pose refine | 0.165 | 0.230 |

## 6.3 QUALITATIVE RESULTS

This section presents qualitative visualizations that illustrate two aspects of the proposed method: (i) image pairs discovered during the cross-domain pairing stage, and (ii) $CAPLR$ improved predictions over the baseline (only synthetic trained network) on real-world samples.

### 6.3.1 CROSS-DOMAIN PAIRS

Pairs discovered by the CDP stage are shown below for the real domains, highlighting semantic and geometric correspondences; rotation and translation differences are also reported.

*A. (Occluded)-LineMOD Pairs*

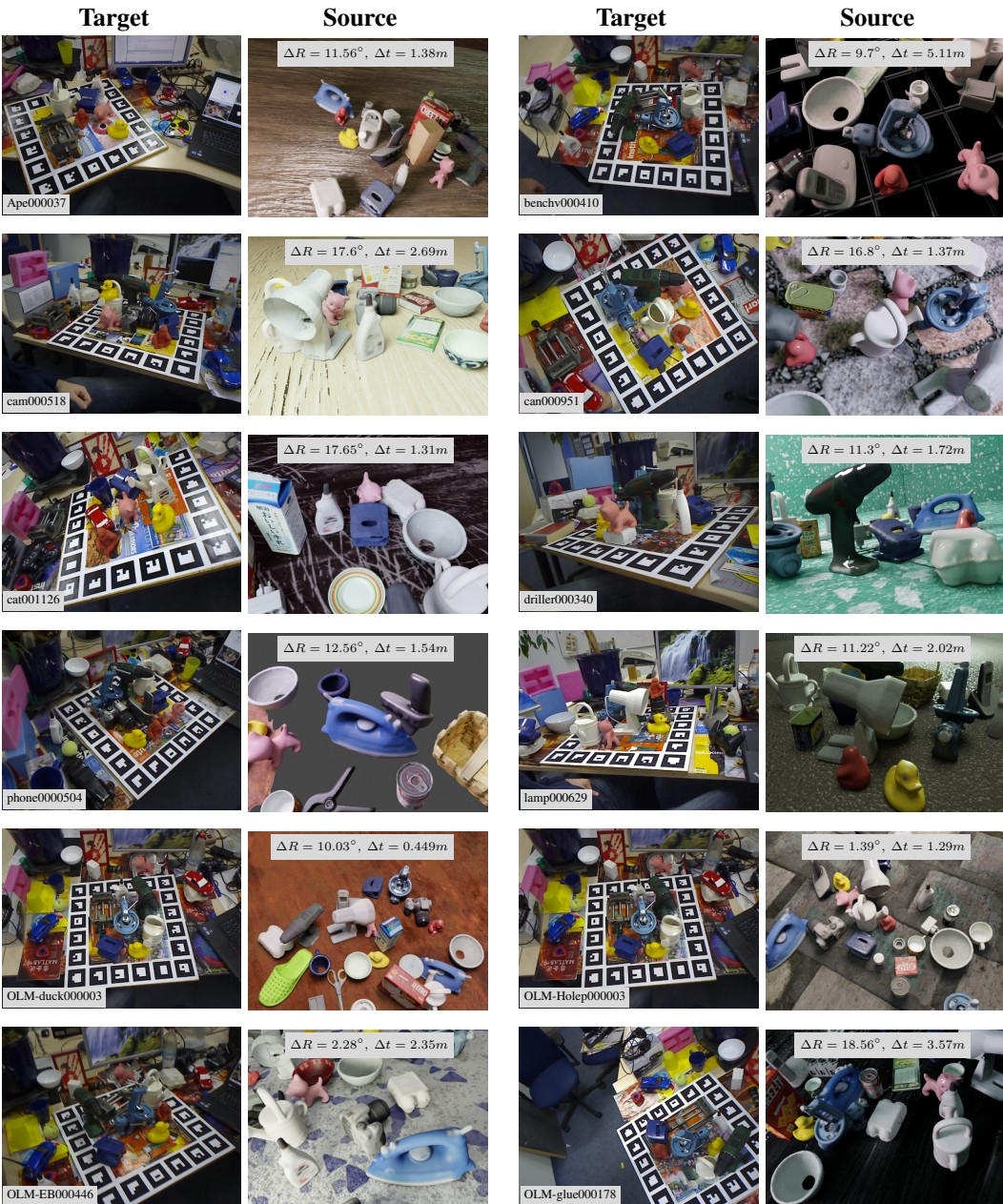

Figure 6: Examples of target-source image pairs in LineMOD (4 first rows) and Occluded-LineMOD (last 2 rows). Rotation and translation differences ($\Delta R$, $\Delta t$) between each pair are indicated.

*B. Lightbox Pairs*

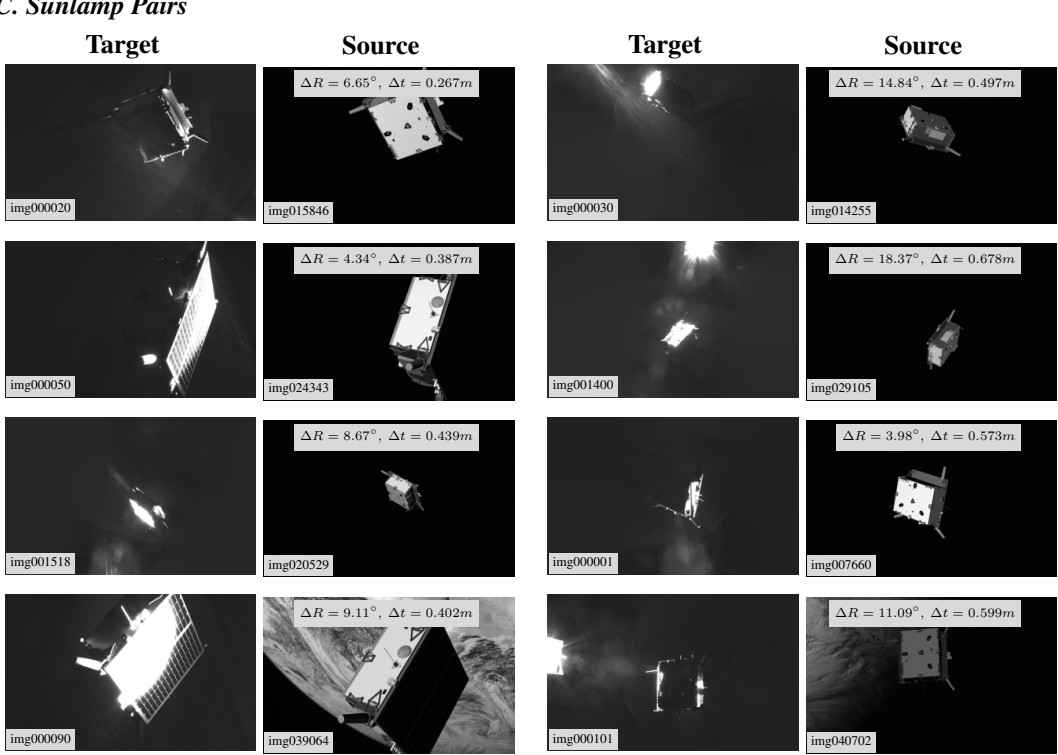

Figure 7: Examples of target-source image pairs for lightbox domain. Rotation and translation differences $(\Delta R, \Delta t)$ between each pair are indicated.

*C. Sunlamp Pairs*

Figure 8: Examples of target-source image pairs from Sunlamp. Rotation and translation differences $(\Delta R, \Delta t)$ between each pair are indicated.

### 6.3.2 PERFORMANCE IMPROVEMENTS OVER BASELINE

We present qualitative comparisons of outputs before adaptation (w/o UDA) and after applying the $CAPLR$ framework, using samples from three target domains: Lightbox, Sunlamp, and LineMOD. In each visualization, the ground-truth mesh or 3D bounding box is overlaid in red, while predictions are shown in blue, facilitating a clear visual assessment of pose alignment accuracy.

***A. LineMOD***

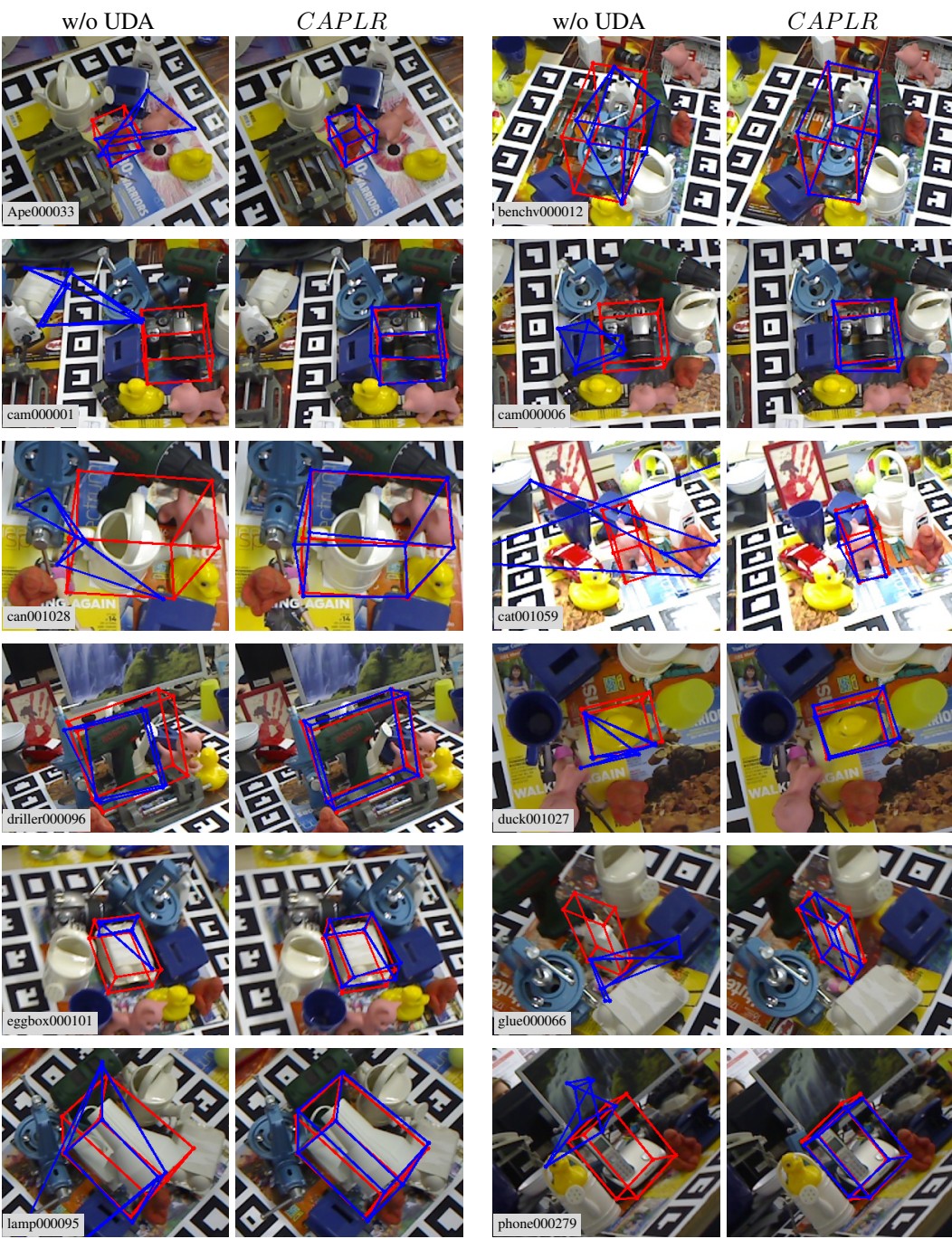

Figure 9: Qualitative comparison of pose predictions on LineMOD images. Each pair shows results without UDA (left) and with $CAPLR$ (right). Ground-truth poses are shown in red, and predictions in blue.

**B. Lightbox**

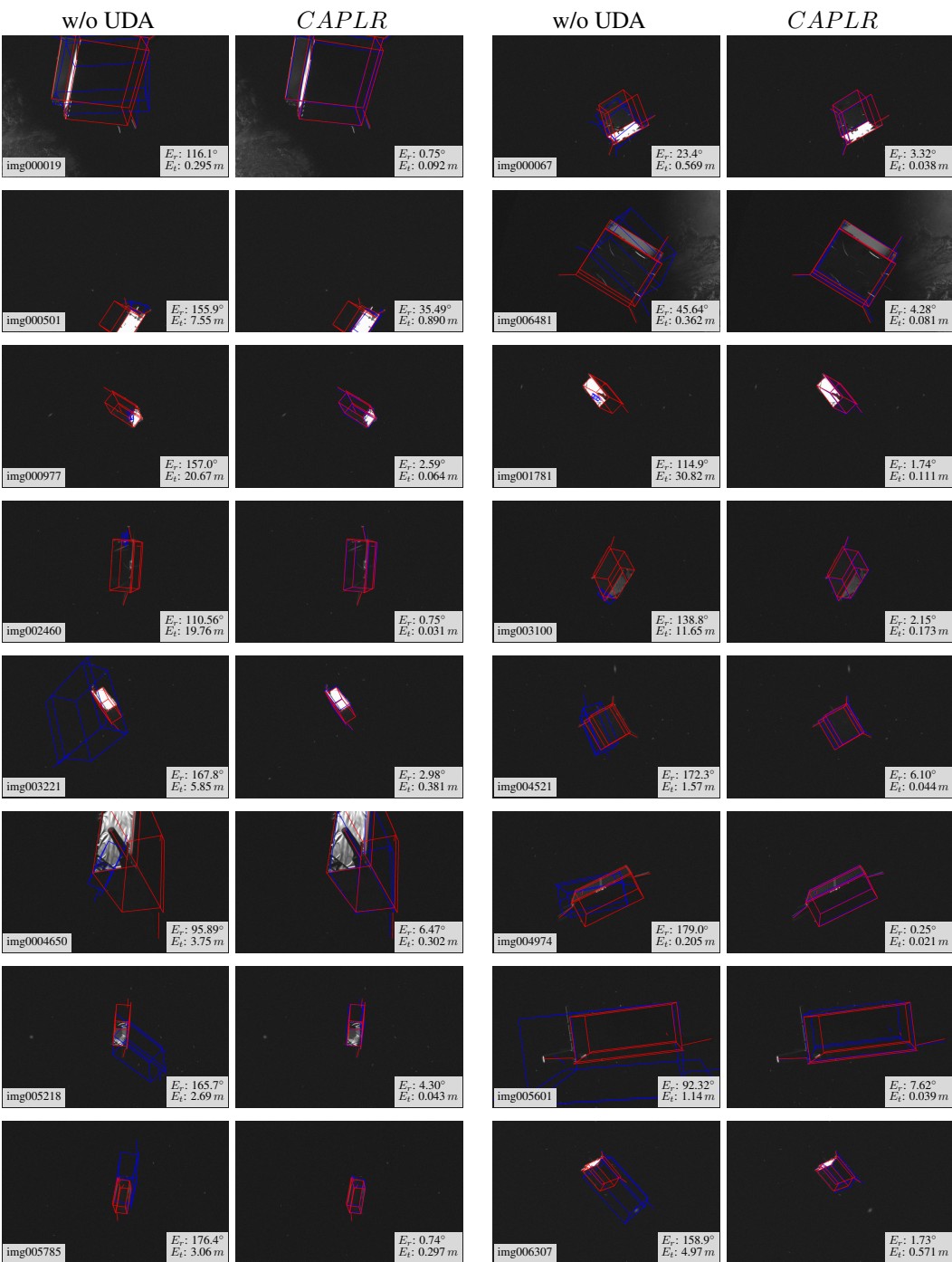

Figure 10: Qualitative comparison of pose predictions on Lightbox images. Each pair shows results without UDA (left) and with $CAPLR$ (right). Rotation and translation errors ($E_r$, $E_t$) are reported. Ground-truth poses are shown in red, and predictions in blue.

## C. Sunlamp

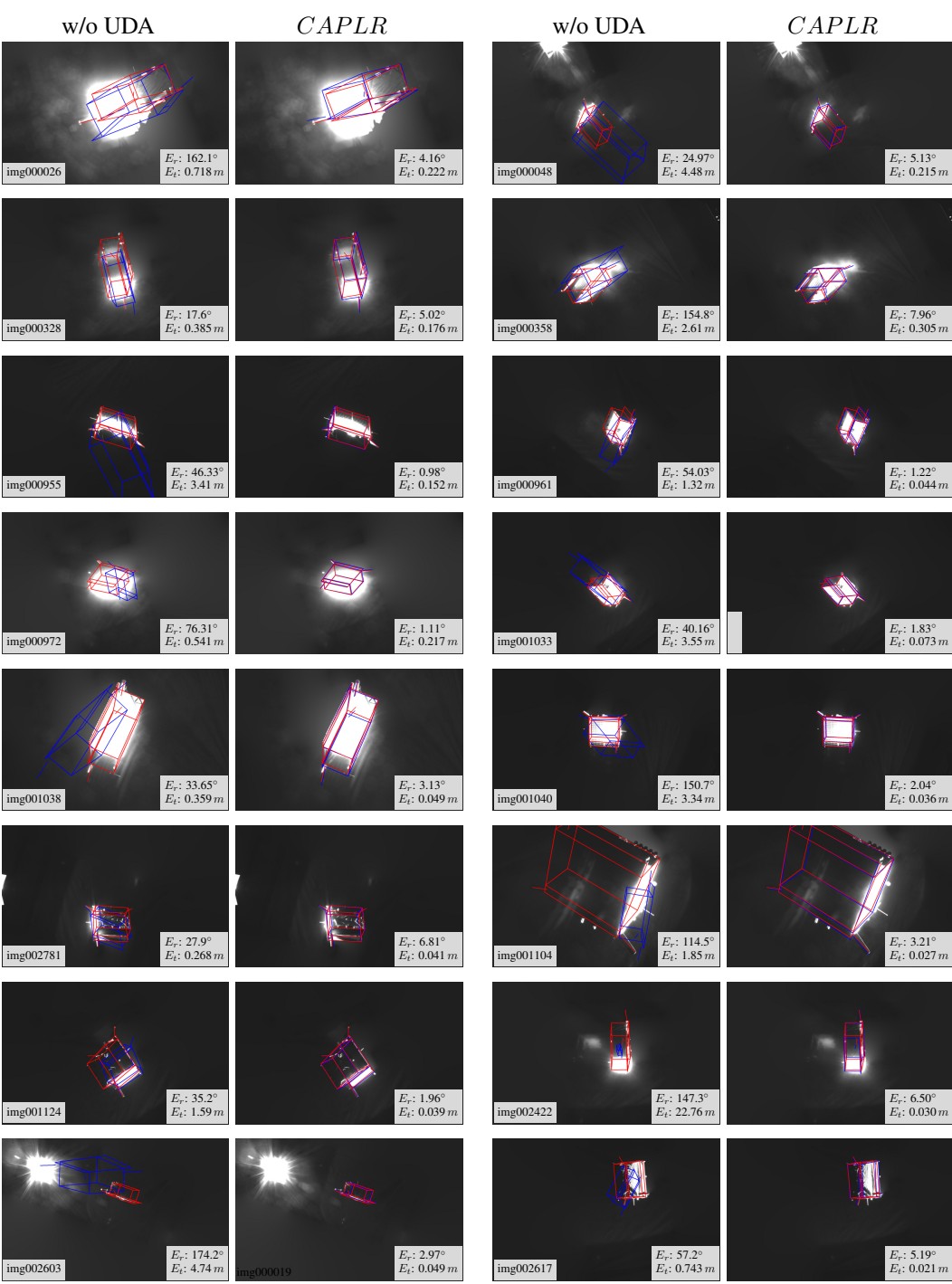

Figure 11: Qualitative comparison of pose predictions on Sunlamp images. Each pair shows results without UDA (left) and with $CAPLR$ (right). Rotation and translation errors ($E_r$, $E_t$) are reported. Ground-truth poses are shown in red, and predictions in blue.

### 6.3.3 FAILURE CASE ANALYSIS OF CROSS-DOMAIN PAIRING

In this section, we analyze the performance of the CDP stage compared to ground-truth pairing. We then examine pairing failure cases, discuss their likely causes, and outline their impact on the overall adaptation performance.

**a) Comparison with Ground-Truth Pairing:**

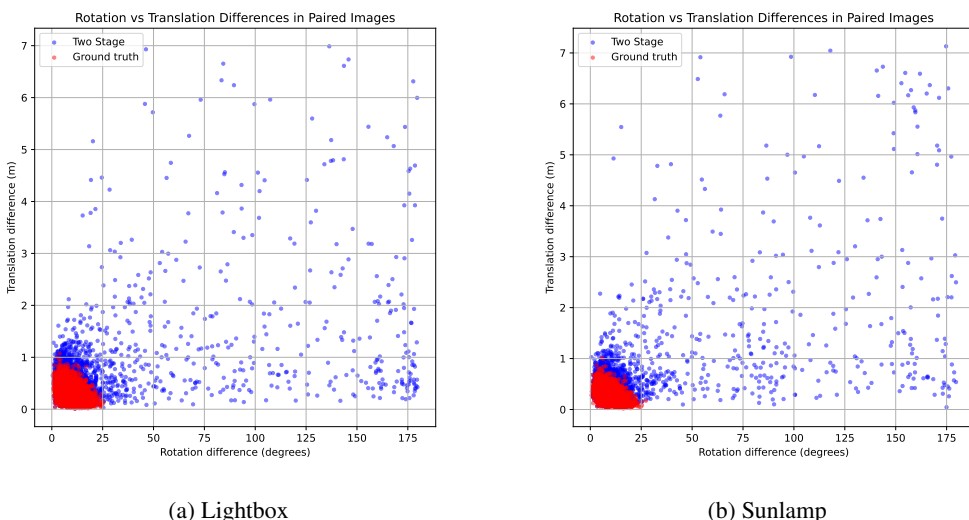

(a) Lightbox           (b) Sunlamp

Figure 12: Comparison of pairing performance in SPEED+. Pairs discovered by the CDP stage are shown in blue, and ground-truth pairs are shown in red. In Lightbox, 90.8% of CDP pairs fall within the low-error range $[0, 1]$ m, $[0, 25]°$, while in Sunlamp, 81.9% of CDP pairs fall in this range.

In Figure 12, we show the distributions of rotational and translational differences between paired samples from the Lightbox and Sunlamp domains of SPEED+. For each domain, we visualize the pairs identified by the CDP stage (blue) alongside the pairs obtained directly from ground-truth labels (red).

On Lightbox, ground-truth pairs lie almost entirely within the region $[0,1]$ m,$[0,25]°$ . The CDP stage exhibits a highly similar structure: 90.8% of CDP-generated pairs fall within this error range, indicating that CDP reliably discovers valid correspondences in this domain. A comparable trend is observed on Sunlamp, where 81.9% of CDP pairs lie within $[0,1]$ m,$[0,25]°$ . These results demonstrate that the CDP stage produces pairings that are strongly aligned with the ground-truth distribution, supporting its effectiveness and robustness across domains, even in real and more challenging illumination settings.

A small fraction of CDP pairs, however, exhibit larger differences in either rotation or translation. Understanding the conditions under which these higher-error pairs occur is important, as they can affect the overall framework performance. We analyze these cases in detail in the following section.

**b) Pairing Failure Analysis:**

In Figure 13, we illustrate several of the worst-discovered pairs from the CDP stage in SPEED+, selected based on large rotational and translational differences. The first three rows correspond to the Sunlamp domain, while the last two rows correspond to the Lightbox domain. For each case, we show the real-synthetic pair, the prediction from the synthetic-only model, and the prediction after CAPLR adaptation.

**Sunlamp:** In the first row, part of the object is occluded by shadows, and a bright lighting source appears at the bottom of the image, creating confusion for the model. As seen in the synthetic-only prediction, this leads to an inaccurate pose estimate. After $CAPLR$, the prediction improves, which is likely because the framework can generalize from better-aligned pairs elsewhere in the dataset,

allowing it to partially correct errors even for poorly paired examples. The second row presents a similarly challenging case with harsh lighting and partially hidden object features, resulting in only a modest improvement after $CAPLR$. In the third row, the real image contains a lighting source that could be confused with the object itself, explaining the pairing failure; nevertheless, $CAPLR$ still provides some correction, plausibly due to the same generalization mechanism.

**Lightbox:** In the fourth row, the real image is very dark, obscuring the object and making it difficult to find a good matching pair. Despite this, $CAPLR$ improves the pose estimate significantly, almost matching the ground-truth, likely by leveraging knowledge learned from other correctly aligned pairs. The fifth row also contains occluded and shadowed objects with limited features, leading to poor initial predictions, but $CAPLR$ still yields noticeable improvement.

Overall, pairing failures occur most frequently under occlusion, harsh shadows, or confusing lighting conditions, which reduce the visibility of key object features. Importantly, $CAPLR$ consistently mitigates the impact of these failures, likely by generalizing from correctly aligned pairs, demonstrating the framework's ability to improve predictions even when individual pairings are imperfect.

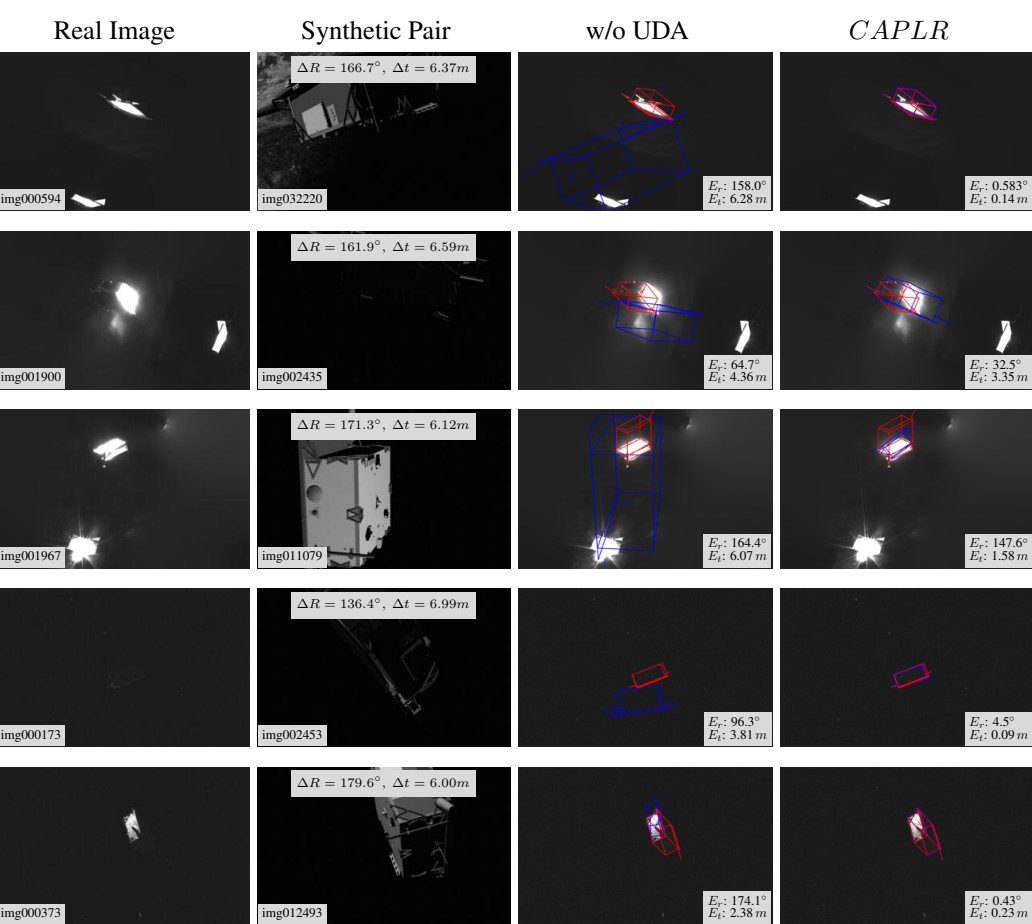

Figure 13: Cross-Domain Pairing failure cases on SPEED+. The first two columns show the discovered real-synthetic pairs along with their rotational and translational differences. The last two columns show the estimated pose on the real image before and after $CAPLR$. Ground-truth poses and predictions are indicated in red and blue, respectively.

### 6.3.4 FAILURE CASES ANALYSIS

Although CAPLR achieves consistent improvements on most samples, there remain cases where the gains are marginal or even where performance degrades. To better understand these limitations, we selected a few representative examples from SPEED+ and conducted a detailed analysis. This examination provides insights into the underlying causes of reduced performance and highlights directions for future research.

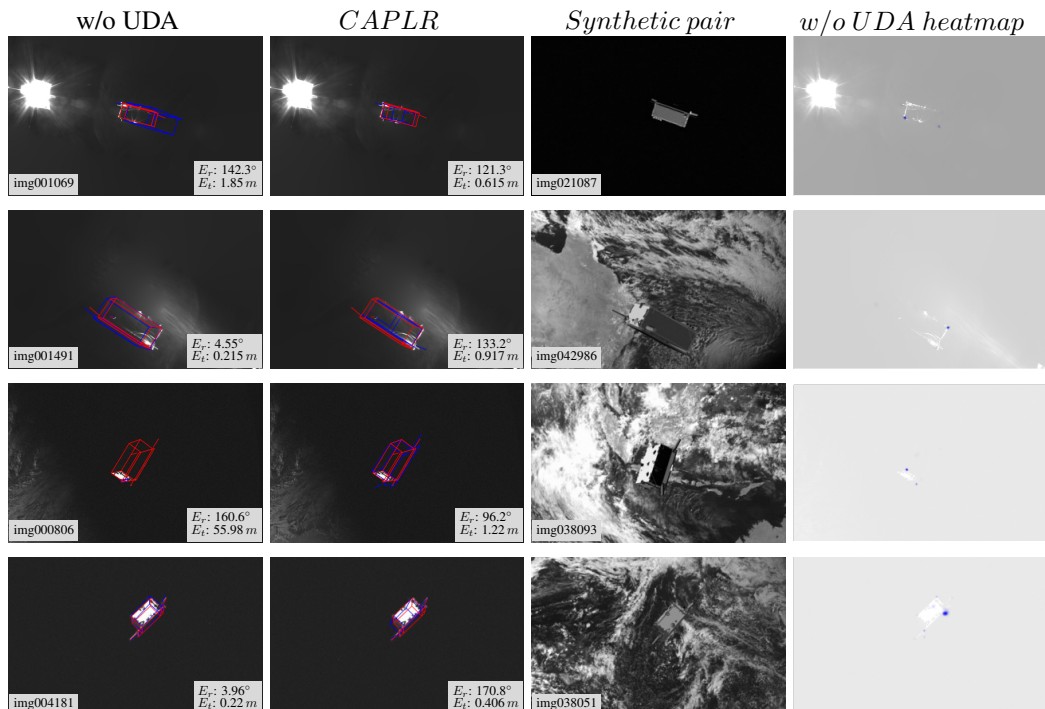

Figure 14: Alignment failure cases on SPEED+. The first two columns show pose estimates without UDA (left) and with $CAPLR$ (right). Ground-truth poses are shown in red and predictions in blue. The third column presents the discovered synthetic pair used for alignment, while the fourth column shows the predictions from the model without UDA that serve as the basis for patch extraction.

**Sunlamp:** In both samples, the discovered image pairs are very similar (in pose) to the target and are generally acceptable. However, the UDA improvements remain limited. For the image in first row, this can be hypothesized due to two main factors. First, the predicted heatmaps fail to highlight all relevant regions in the target image, leading to inaccurate patch extraction that misses important cues. This issue is likely associated to the harsh lighting, which obscure object details leading to confusion in the predictions. Second, the object's orientation in the image exposes only a single plane with relatively few discriminative features, further reducing the effectiveness of alignment.

For the image in second row, the degradation is more pronounced. In addition to the aforementioned heatmap limitation, the presence of the Earth background in the synthetic image appears to mislead the patch-based alignment, since the corresponding real image only contains a dark background as well as affected by the scattered light around the keypoint regions. This discrepancy pushes the alignment towards unreliable feature correspondences.

**Lightbox:** For the third row, the predicted heatmaps successfully highlight relevant regions, enabling accurate patch localization. Nevertheless, the improvement in rotation accuracy remains modest. This is likely due to background differences between the synthetic and real images, combined with the fact that the real image appears significantly darker, limiting the availability of prominent object features.

For the fourth row, the results again show degradation, despite the heatmaps highlighting appropriate regions. Similar to the second row, this behavior can be explained by the influence of backgrounds in the patch between the synthetic and real images, which undermines the alignment quality.

