# OpenReview forum: "Unsupervised Domain Adaptation for 6-DoF Pose Estimation with Contrastive Alignment and Pseudo-Label Refinement"
_ICLR.cc/2026/Conference — Submitted to ICLR 2026_

### Official Review · Reviewer_jkYC · 2025-10-25

**Soundness:** 2
**Presentation:** 2
**Contribution:** 3
**Rating:** 4
**Confidence:** 4

**Summary:**

The paper proposes CAPLR, a keypoint-based unsupervised domain adaptation (UDA) framework for 6-DoF pose estimation. CAPLR addresses synthetic-to-real domain gaps through three key components: Cross-Domain Pairing (CDP) for identifying pose-consistent source-target image pairs, Local Patch Contrastive Alignment (LPCA) for dual-level (backbone and head) feature alignment, and Consistency-Based Pseudo-Label Refinement (CBPR) for stabilizing predictions via augmented views. Experiments on LineMOD, Occluded-LineMOD, HomeBrewedDB, and SPEED+ datasets show state-of-the-art performance.

**Strengths:**

+ An interesting aspect of the method is its two-step, cross-domain pairing strategy, where initial anchors derived from heatmap activations are subsequently refined using pose errors. The ablation study in Table 5 also verifies the effectiveness of this strategy.

+ The pseudo-label refinement strategy leveraging spatial distance and confidence effectively identifies more reliable keypoint pseudo-labels.

**Weaknesses:**

-- The comparative experiment settings on SPEED+ may be unfair. Did the other methods use all the test data for domain adaptation? This should be explicitly labeled in the table, and it should be ensured that the other methods also use the identical training and testing setup.

-- The evaluation is limited by the absence of two key comparisons for the proposed method: one trained only on synthetic data and one trained on annotated synthetic and real data. This omission makes it difficult to assess lower and upper bounds of the method's performance.

-- The method heavily relies on keypoint predictions and their corresponding heatmaps. For symmetric objects where reliable keypoints cannot be obtained, or in cluttered environments where background objects with similar appearances lead to false positive keypoints, the method may fail. To understand whether the method exhibits this issue, additional experiments on datasets like T-LESS could be beneficial.

-- It would be better to move key ablations to the main paper rather than in the appendix.

Minor

-- Why does the actual reference entry for Self6D++ correspond to GDR-Net?

-- The reference entry for PoseCNN is duplicated.

-- L320-323: Missing commas in these sentences.

**Questions:**

-- Does this keypoint-based method include a 2D object detection module? If not, how does it handle multiple instances of the same object within a single image?

---

> ### Author Response · Authors · 2025-11-20
> **Response to Reviewer jkYC (w1: Comparisons on SPEED+)**
>
> ###
> We thank the reviewer for the constructive feedback. We address the concerns below and have uploaded a revised version of the paper reflecting all reviewers comments.
>
> ---
> > **Comment:** The comparative experiment settings on SPEED+ may be unfair. Did the other methods use all the test data for domain adaptation? This should be explicitly labeled in the table, and it should be ensured that the other methods also use the identical training and testing setup.
>
> ---
> **Response:** We acknowledge that some of the reported methods were trained under different setups. In particular, YOLOv8s[1] and UAKD[2] were trained exclusively on synthetic images, as these works primarily focused on efficiency and deployability. PVSPE[3] also relied solely on synthetic data, but it uses a substantially more complex, multi-task learning framework with a heavier architecture. FA-VAE[4] was similarly trained only on synthetic images.
>
> In contrast, the remaining four methods use a combination of synthetic and unlabeled real images, consistent with
> our setting. We have clarified these differences in both the revised Table-3 and the accompanying text in Section 4.2-C.
>
> ---
> **References:**
>
> [1] Bechini and Lavagna. Robust and efficient single-cnn-based spacecraft relative pose estimation from monocular images. Acta Astronautica, 2025.
>
> [2] Ali Ousalah et al. Uncertainty-aware knowledge distillation for compact and efficient 6dof pose estimation. IROS, 2025
>
> [3] Yanget al.  Pvspe: A pyramid vision multitask transformer network for spacecraft pose estimation. Advances in Space
> Research,2024b.
>
> [4] Yanfang et al. Feature-aided pose estimation approach based on variational auto-encoder structure for spacecrafts. Chinese Journal of Aeronautics, 2024

---

> > ### Comment · Reviewer_jkYC · 2025-11-24
> >
> > Is there any overlap between the unlabeled real images used for training and the test set? This should be explicitly stated.

---

> ### Author Response · Authors · 2025-11-20
> **Response to Reviewer jkYC (w2: Upper-lower bounds comparison)**
>
> ---
> > **Comment:** The evaluation is limited by the absence of two key comparisons for the proposed method: one trained only on synthetic data and one trained on annotated synthetic and real data. This omission makes it difficult to assess lower and upper bounds of the method's performance.
>
> ---
> ###
> **Response:**
>
> Following your suggestion, which was also raised by Reviewer **Givv** , we have added results comparing the model trained only on synthetic images and the model trained on synthetic + real images in the revised manuscript (supplementary Section-6.1.4). Here, we provide a summary of these results.
>
> The lower bound corresponds to training on synthetic images only, while the upper bound corresponds to training on synthetic images and fine-tuning on real images for the same number of alignment epochs as in CAPLR. For LineMOD and Occluded-LineMOD, the fine-tuning uses the same real images as those available for adaptation; for SPEED+, we use the first 20\% of real images
>
> As shown in Tables below, CAPLR achieves consistent and substantial gains over the synthetic-only baseline. In most cases, the improvement from synthetic-only → CAPLR exceeds the difference between CAPLR and the upper-bound (synthetic + real) model, demonstrating the effectiveness of CAPLR in closing the performance gap toward real data while using minimal real images.
>
> ---
> **Tables:**
>
> **Table: Results on LineMOD – Baseline (S), CAPLR (S+R⁻), and Real-Finetuned Model (S+R)**
>
> | Method      | Data   | Ape  | Bvise | Cam  | Can  | Cat  | Drill | Duck | Eggbox | Glue | Holep | Iron | Lamp | Phone | **Mean** ↑ |
> |------------|--------|------|-------|------|------|------|-------|------|--------|------|-------|------|------|-------|------------|
> | **CAPLR_s**| S      | 49.7 | 94.5  | 74.1 | 93.3 | 95.1 | 94.3  | 78.3 | 96.9   | 84.2 | 91.1  | 98.7 | 63.8 | 82.8  | 84.4       |
> |            | S+R⁻   | 82.1 | 95.2  | 93.2 | 97.4 | 98.8 | 97.0  | 89.5 | 100.0  | 96.9 | 95.1  | 98.8 | 78.0 | 93.7  | 93.5       |
> |            | S+R    | 95.1 | 98.9  | 95.1 | 95.8 | 93.1 | 99.7  | 96.3 | 100    | 96.1 | 98.7  | 99.8 | 86.6 | 96.8  | 96.3       |
> | **CAPLR_m**| S      | 75.2 | 99.5  | 85.1 | 95.6 | 96.6 | 95.1  | 96.0 | 99.8   | 84.6 | 92.9  | 99.7 | 75.2 | 90.9  | 90.8       |
> |            | S+R⁻   | 90.7 | 99.5  | 93.2 | 97.9 | 99.3 | 97.7  | 98.8 | 100.0  | 99.5 | 97.0  | 99.9 | 94.1 | 96.6  | 97.2       |
> |            | S+R    | 98.1 | 99.9  | 97.7 | 98.4 | 100  | 99.9  | 99.9 | 100    | 98.6 | 99.7  | 100  | 96.8 | 99.9  | 99.2       |
>
>
>
> **Table: Results on Occluded-LineMOD – Baseline (S), CAPLR (S+R⁻), and Real-Finetuned Model (S+R)**
>
> | Method      | Data   | Ape  | Can  | Cat  | Drill | Duck | Eggbox | Glue | Holep | **Mean** ↑ |
> |------------|--------|------|------|------|-------|------|--------|------|-------|------------|
> | **CAPLR_s**| S      | 54.6 | 80.3 | 56.1 | 79.6  | 55.8 | 91.1   | 62.9 | 80.3  | 70.3       |
> |            | S+R⁻   | 62.0 | 91.3 | 61.2 | 86.1  | 69.4 | 95.6   | 75.6 | 93.6  | 79.3       |
> |            | S+R    | 85.0 | 95.8 | 87.7 | 90.3  | 82.9 | 95.3   | 80.6 | 98.4  | 89.5       |
> | **CAPLR_m**| S      | 54.6 | 84.0 | 54.5 | 92.5  | 79.5 | 97.4   | 68.6 | 86.7  | 77.2       |
> |            | S+R⁻   | 67.0 | 93.9 | 71.3 | 95.0  | 83.3 | 98.0   | 85.8 | 92.4  | 85.8       |
> |            | S+R    | 90.0 | 97.4 | 95.1 | 96.8  | 88.4 | 98.4   | 87.9 | 99.0  | 94.1       |
>
>
> **Table: SPEED+ – Baseline (S), CAPLR (S+R⁻), and Real-Finetuned Model (S+R)**
>
> | Method      | Data | Lightbox  $E_T / E_T^{norm}$ ↓ | Lightbox  $E_R$ ↓ | Lightbox $E_{pose}$ ↓ | Sunlamp $E_T / E_T^{norm}$ ↓ | Sunlamp $E_R$ ↓ | Sunlamp $E_{pose}$ ↓ | Mean $E_{pose}$ ↓ |
> |------------|------|-------------------------------|-----------------|----------------------|--------------------------------|----------------|---------------------|-----------------|
> | **CAPLR_s**| S    | 0.221 / 0.036                 | 8.98            | 0.193                | 0.426 / 0.065                  | 18.93          | 0.395               | 0.294           |
> |            | S+R⁻ | 0.173 / 0.029                 | 7.80            | 0.165                | 0.215 / 0.035                  | 11.17          | 0.230               | 0.198           |
> |            | S+R  | 0.127 / 0.020                 | 4.74            | 0.103                | 0.131 / 0.021                  | 5.38           | 0.115               | 0.109           |
> | **CAPLR_m**| S    | 0.162 / 0.026                 | 5.31            | 0.119                | 0.288 / 0.044                  | 11.09          | 0.237               | 0.178           |
> |            | S+R⁻ | 0.103 / 0.018                 | 3.54            | 0.080                | 0.126 / 0.022                  | 5.77           | 0.122               | 0.101           |
> |            | S+R  | 0.097 / 0.016                 | 2.99            | 0.068                | 0.106 / 0.017                  | 3.69           | 0.082               | 0.075           |

---

> ### Author Response · Authors · 2025-11-20
> **Response to Reviewer jkYC (w3: Symmetric Objects)**
>
> ---
> > **Comment:** The method heavily relies on keypoint predictions and their corresponding heatmaps. For symmetric objects where reliable keypoints cannot be obtained, or in cluttered environments where background objects with similar appearances lead to false positive keypoints, the method may fail. To understand whether the method exhibits this issue, additional experiments on datasets like T-LESS could be beneficial
>
> ---
> ###
> **Response:**
>
> The framework remains valid for symmetric objects. While it is true that the model may confuse keypoints, the extracted
> features will still be aligned with their symmetric counterparts. This works in our case because the features of symmetric
> keypoints are very similar, so the alignment is effective regardless of which keypoint matches which.
> This is supported by the observed improvements on the two symmetric objects (Eggbox and Glue) present in both
> LineMOD (Table1) and Occluded-LineMOD (Table2) datasets. In the revised article (lines 351-352), we clarified that these datasets include
> symmetric objects, as this detail may not have been clear to readers. Following prior work, the improvement is evaluated using the ADD-S metric, which accommodates symmetric ambiguities.
>
> Evaluating on textureless objects, such as those in T-LESS, introduces a different challenge of localizing objects
> without texture, indeed a challenging problem. Currently, we regard this as beyond the scope of this study.

---

> > ### Comment · Reviewer_jkYC · 2025-11-24
> >
> > T-LESS is a representative dataset featuring objects with similar appearances and symmetric shapes, making it essential for evaluating keypoint-based methods. This aspect should not be overlooked. As for object localization or detection, numerous well-established 2D detectors can be employed if the authors' approach does not include 2D detection.

---

> ### Author Response · Authors · 2025-11-20
> **Response to Reviewer jkYC (w4: Abaltion study in the main paper)**
>
> ---
> > **Comment:** It would be better to move key ablations to the main paper rather than in the appendix.
>
> ---
> ###
> **Response:**
>
> We acknowledge that ablation studies play a crucial role in assessing each component of our method and in offering a
> comprehensive understanding of the approach. The original submission could not include them due to ICLR’s 9-page
> limit. In the revised version, taking advantage of the 10-page allowance for rebuttal and camera-ready submissions (as
> per author guidelines), we have added a new page and incorporated the ablation studies into the main paper

---

> ### Author Response · Authors · 2025-11-20
> **Response to Reviewer jkYC (Minor weaknesses)**
>
> ###
> **Response:**
>
> We appreciate the reviewer’s careful reading and valuable feedback. Your minor remarks have been taken into consideration and corrected in the revised version of the paper.

---

> ### Author Response · Authors · 2025-11-20
> **Response to Reviewer jkYC (Q: Handling multiple instances of the same object)**
>
> ---
> > **Question:** Does this keypoint-based method include a 2D object detection module? If not, how does it handle multiple instances of the same object within a single image?
>
> ---
> ###
> **Response:**
>
> Our current framework, similar to existing UDA methods we compare against, is designed to handle a single instance
> of the target object per image. Nevertheless, extending it to scenarios with multiple instances is both interesting and
> feasible. We outline two potential strategies for achieving this extension.
>
> ### 1. Object pre-extraction
> A straightforward solution involves introducing an object detection or segmentation module at the beginning of the pipeline. Each detected object instance can then be processed independently using our proposed framework, as if it were a single-instance image. This modular approach allows the system to scale naturally to scenes with multiple objects while keeping the main method unchanged.
>
> ### 2. End-to-end multi-instance extension
> Alternatively, we can avoid external detectors by modifying the network to predict multiple poses (one for each visible instance). In this setup, the pairing strategy would operate at the instance level: each real image instance would be matched with a synthetic image containing an object of a similar pose. During feature alignment, the contrastive loss would be computed only between corresponding instance patches, and the total loss would be obtained by summing over all instances.
>
> For example, if a real image contains three instances, the model would pair each of them with a distinct synthetic image showing a similar pose. The alignment loss would be computed instance-wise and then aggregated across all three.
>
> During pseudo-label refinement, if the network outputs multiple sets of keypoints (e.g., 3×K keypoints for three objects), an additional matching step is required to associate keypoints with their respective objects. Since all keypoints originate from the same image, spatial proximity can be used to cluster them by instance before applying the refinement stage as usual.
>
>
> ---
> Finally, we hope our responses have clarified your concerns, and we remain open to any further feedback or questions you may have.

---

> ### Author Response · Authors · 2025-11-25
> **Response to Reviewer jkYC (Q: Adaptation-Test overlap in SPEED+)**
>
> ---
> > **Question:** Is there any overlap between the unlabeled real images used for training and the test set? This should be explicitly stated?
>
> ---
> ###
> **Response:**
>
> For SPEED+, as noted in lines 363–364, we follow prior work (SPNv2[3], Lava1302 [4], VPU [5] and TangoUnchained) by using all test images without labels for adaptation. We acknowledge that this adaptation–test overlap can raise concerns about the method’s validity. However, our approach has already been validated on multiple other datasets (LineMOD, Occluded-LineMOD, HoebrewedDB) where adaptation images and test images are fully disjoint, demonstrating that the method does not rely on overlap to perform well.
>
> To further address scenarios with limited real data, we conducted an ablation on SPEED+ (Table 6) in which only 10% and 20% of the real images were used for adaptation while testing was performed on the full test set. The performance remains strong under these reduced-data settings, confirming that the method is not dependent on extensive overlap.
>
> Finally, to remove any remaining concerns regarding adaptation–test overlap, we evaluated the SPEED+ models on the recent SHIRT dataset. SHIRT contains two image sequences (roe1, roe2) exhibiting lighting characteristics similar to the SPEED+ lightbox domain, making it an appropriate out-of-distribution benchmark. We report results comparing:
>
> (1) a model trained only on synthetic data,
>
> (2) a model trained on synthetic data and adapted on SPEED+ real lightbox images,
>
> (3) a model trained on synthetic + real data, and
>
> (4) the approaches from [1] and [2], which use similar architectures trained on synthetic data with additional sequence-filtering mechanisms.
>
> Our method outperforms both [1] and [2] and shows a clear reduction in the domain gap, approaching real-data performance, even though SHIRT was never used during adaptation. This further confirms that our approach generalizes beyond the SPEED+ domain and is not reliant on adaptation–test image overlap.
>
>
>
> **Table: Generalization to the SHIRT dataset**
> Metrics are reported for two sequences (roe1 and roe2). Lower is better (↓).
>
> | Method                | Data   | roe1 $E_T$ [m] ↓ | roe1 $E_R$ [°] ↓ | roe1 $E_pose$ ↓ | roe2 $E_T$ [m] ↓ | roe2 $E_R$ [°] ↓ | roe2 $E_pose$ ↓ |
> |-----------------------|--------|-----------------|-----------------|----------------|-----------------|-----------------|----------------|
> | Cassinis et al., 2023 [1] | S + A  | 0.250 ± 0.06    | 14.00 ± 11.00   | ---            | 0.380 ± 0.06    | 9.00 ± 7.50     | ---            |
> | Park et al., 2023 [2]     | S + B  | 0.303 ± 0.35    | 17.17 ± 42.25   | ---            | 0.203 ± 0.23    | 4.67 ± 14.90    | ---            |
> | **CAPLR_m**            | S      | 0.302 ± 1.71    | 10.76 ± 27.76   | 0.227 ± 0.60   | 0.109 ± 0.29    | 5.98 ± 22.00    | 0.125 ± 0.41   |
> |                        | S + R⁻ | 0.250 ± 0.86    | 10.70 ± 27.95   | 0.219 ± 0.55   | 0.101 ± 0.17    | 4.22 ± 17.71    | 0.092 ± 0.33   |
> |                        | S + R  | 0.307 ± 2.45    | 10.27 ± 27.63   | 0.219 ± 0.63   | 0.090 ± 0.41    | 3.79 ± 14.44    | 0.083 ± 0.29   |
>
> **Notes:**
> - **A:** Novel lighting augmentation pipeline + adaptive Unscented Kalman Filter
> - **B:** Multitask learning + Unscented Kalman Filter
>
>
>
> **References**
>
> [1] Cassinis, Lorenzo Pasqualetto, et al. "Leveraging neural network uncertainty in adaptive unscented Kalman Filter for spacecraft pose estimation." Advances in Space Research 71.12 (2023): 5061-5082.
>
> [2] Park, Tae Ha, and Simone D’Amico. "Adaptive neural-network-based unscented kalman filter for robust pose tracking of noncooperative spacecraft." Journal of Guidance, Control, and Dynamics 46.9 (2023): 1671-1688.
>
> [3] Park, Tae Ha, and Simone D’Amico. "Robust multi-task learning and online refinement for spacecraft pose estimation across domain gap." Advances in Space Research 73.11 (2024): 5726-5740.
>
> [4] Zi Wang, Minglin Chen, Yulan Guo, Zhang Li, and Qifeng Yu. Bridging the domain gap in satellite
> pose estimation: A self-training approach based on geometrical constraints. IEEE transactions
> on aerospace and electronic systems, 60(3):2500–2514, 2023.
>
> [5] Juan Ignacio Bravo P´erez-Villar, ´Alvaro Garc´ıa-Mart´ın, and Jes´us Besc´os. Spacecraft pose estima-
> tion based on unsupervised domain adaptation and on a 3d-guided loss combination. In European
> Conference on Computer Vision, pp. 37–52. Springer, 2022.

---

> > ### Comment · Reviewer_jkYC · 2025-11-28
> >
> > I appreciate the new results on the SHIRT dataset, which effectively resolve my concerns regarding the training-testing overlap. I strongly recommend replacing the SPEED+ experiment in the main paper with these new results to ensure a fair evaluation and avoid potentially misleading readers.

---

> ### Author Response · Authors · 2025-11-25
> **Response to Reviewer jkYC (Q: Evaluation on T-LESS Dataset)**
>
> ---
> > **Question:** T-LESS is a representative dataset featuring objects with similar appearances and symmetric shapes, making it essential for evaluating keypoint-based methods. This aspect should not be overlooked. As for object localization or detection, numerous well-established 2D detectors can be employed if the authors' approach does not include 2D detection.
>
> ---
> ###
> **Response:**
>
> We acknowledge the reviewer’s request in evaluating the algorithm on additional dataset T-LESS. Within the remaining rebuttal period, we will do our best to perform the evaluation on T-LESS dataset and provide either full or partial results (depending on time constraints) to strengthen our arguments.

---

> ### Author Response · Authors · 2025-12-01
> **Response to Reviewer jkYC (Incorporation of SHIRT Results into the Main Paper)**
>
> > **Comment:** I appreciate the new results on the SHIRT dataset, which effectively resolve my concerns regarding the training-testing overlap. I strongly recommend replacing the SPEED+ experiment in the main paper with these new results to ensure a fair evaluation and avoid potentially misleading readers.
>
> **Response**
>
> Following the reviewer’s suggestion, we included the results on the SHIRT dataset to the generalization experiments of CAPLR (Table 4 in the main paper). We now report generalization on two datasets: models adapted on LineMOD and tested on HomebrewedDB, and models adapted on Lightbox from SPEED+ and tested on SHIRT.
>
> We kept the SPEED+ results along with SHIRT, because SPEED+ is a valuable dataset for our main paper, especially since it serves as the primary source for most of our ablation studies. However, we added clarifications to the adaptation setup in the experiments (i.e., using 100\% of the real images for adaptation to ensure fair comparison with prior work) and to avoid any possible misunderstanding by the readers (lines 363-364, lines 430-454-455).

---

> ### Author Response · Authors · 2025-12-01
> **Response to Reviewer jkYC (Partial Results on T-LESS Dataset)**
>
> > **Comment:** T-LESS is a representative dataset featuring objects with similar appearances and symmetric shapes, making it essential for evaluating keypoint-based methods. This aspect should not be overlooked. As for object localization or detection, numerous well-established 2D detectors can be employed if the authors' approach does not include 2D detection.
>
> **Response**
>
> Given the limited remaining rebuttal period, we conducted experiments on Sequence 4 of the T-LESS dataset, which we selected because it contains only four objects and one instance per object, making it feasible to run with our current implementation. Following the official BOP split, we used 10\% of real images for adaptation and evaluated on remaining 90\%.
>
> The table below reports the ADD-S scores for three settings:
>
> - Synthetic-only trained model (S)
> - CAPLR (S+R-)
> - Synthetic+real trained model (S+R).
>
> Although ADD-S is not the standard metric for T-LESS, we adopt it to remain consistent with our LineMOD evaluation, especially to show the performance inprovements compared to synthetic only models. Our primary objective is to verify whether CAPLR remains valid and effective for highly symmetric objects (as suggested by the Reviewer). Note that the UDA methods compared in the main paper (such as ONDA-pose[1], TexPose[2], SMOCNet[3], and MAST[4]) do not include the evaluation of the T-LESS dataset. The proposed CAPLR approach already performs better on the datasets they do evaluate on, including LineMOD, Occluded LineMOD, and HomebrewedDB.
>
> Importantly, T-LESS provides a more challenging dataset because of its numerous symmetric objects, whereas LineMOD contains only two symmetric objects. The results on T-LESS Sequence 4 clearly demonstrate that CAPLR consistently improves performance on these symmetric objects, supporting the reviewer’s request for stronger validation of CAPLR’s generalization capabilities.
>
> **Table: Results on TLESS-SEQUENCE4**
>
> | Method | Data   | Object5 | Object8 | Object26 | Object28 | **Mean** ↑ |
> |--------|--------|---------|---------|----------|----------|------------|
> | **$CAPLR_m$** | S       | 64.2 | 65.8 | 73.1 | 42.2 | 61.3 |
> |              | S+R- | 78.9 | 82.8 | 91.2 | 50.0 | 75.7 |
> |              | S+R     | 98.9 | 94.7 | 99.8 | 47.9 | 85.3 |
>
>
> **References**
>
> [1] Tao Tan and Qiulei Dong. Onda-pose: Occlusion-aware neural domain adaptation for self-supervised 6d object pose estimation. In Proceedings of the Computer Vision and Pattern Recognition Conference, pp. 16829–16838, 2025.
>
> [2] Hanzhi Chen, Fabian Manhardt, Nassir Navab, and Benjamin Busam. Texpose: Neural texture learning for self-supervised 6d object pose estimation. In Proceedings of the IEEE/CVF Conference on Computer Vision and Pattern Recognition, pp. 4841–4852, 2023.
>
> [3] Tao Tan and Qiulei Dong. Smoc-net: leveraging camera pose for self-supervised monocular object pose estimation. In Proceedings of the IEEE/CVF Conference on Computer Vision and Pattern Recognition, pp. 21307–21316, 2023.
>
> [4] Yichen Zhang, Jiehong Lin, Ke Chen, Zelin Xu, Yaowei Wang, and Kui Jia. Manifold-aware self-training for unsupervised domain adaptation on regressing 6d object pose. In Proceedings of the Thirty-Second International Joint Conference on Artificial Intelligence, pp. 1740–1748, 2023.

---

### Official Review · Reviewer_LE1D · 2025-10-31

**Soundness:** 2
**Presentation:** 2
**Contribution:** 1
**Rating:** 2
**Confidence:** 2

**Summary:**

This paper proposes an unsupervised domain adaptation algorithm for the 6-DoF object pose estimation task. Given (source) synthetic datasets having annotations and (target) real-world data with small scale data, the task needs to reduce the domain gap and predict the accurate objects' pose information.

This paper proposes (1) Cross-domain pairing, (2) contrastive alignment, and (3) consistency based pseudo label refinement. The proposed schemes are to reduce the domain gap in embedding spaces as well as objects' pose space.

Overall, the performance looks good compared to previous studies, but there is no ablation study in the manuscript.

**Strengths:**

Overall flow of this paper is readable and understandable. The proposed strategies are aligned with the unsupervised domain adaptation using image/pose pairing in source-target data (Sec. 3.2.1) and embedding alignment (Sec. 3.2.2).  Nonetheless it is not that convinced that the proposed methods are novel and unique.

**Weaknesses:**

__W1. No ablation study in the manuscript?__
While the authors propose two dominant techniques in the manuscript, but I cannot find the ablation study within the main paper. Surprisingly, the ablation studies were found in the supplementary, but I am a bit worried about such a paper writing. While it is good for me to read the manuscript, but it needs to be more compact to make more margins and locate the ablation studies in the manuscript. I personally guess that the authors should have spent lots of efforts in paper writing. The current writing is not good enough for the submission. It needs reformulation.

__W2. Novelty__
Even though the authors present the background knowledge for the concept of the embedding alignments in Lines 216-247, it is not something new. More technical speaking, it is simply align the two embedding spaces, one for the embedding vectors from backbone network and the other for the objects' poses from the header network, from the target domain and the source domain. Even, the authors simply leverage the InfoNCE loss. Can the authors provide more clues or claims why the proposed schemes are novel?

Moreover, it is not also something new to introduce pseudo labeling in the unsupervised domain adaptation problem. There are lots of related works who also use the same idea, but based on the submission, I cannot clearly tell the differences.

**Questions:**

I do not have the specific questions. Instead, I hope the authors to answer to my questions writing in the weakness section.

Overall, it is easy to read the paper, but it needs to be more concise and compact. Ablation studies are not written in the manuscript, but the supplementary material. I am not sure why the Section 3.1 is necessary. The paper needs lots of revision.

Beside from the writing, I do not think that the authors present the novel ideas or contributions. There are lots of existing strategy to solve the unsupervised domain adaptation problem for the 6-DoF pose estimation problem. However, I cannot clearly tell the differences between the previous works and the proposed schemes.

---

> ### Author Response · Authors · 2025-11-20
> **Response to Reviewer LE1D (w1: Abaltion study in the main paper)**
>
> We thank the reviewer for the feedback on our Work. We address the concerns below and have uploaded a revised version of the paper reflecting all reviewers comments.
>
> ---
> > **Comment** While the authors propose two dominant techniques in the manuscript, but I cannot find the ablation study within the main paper. Surprisingly, the ablation studies were found in the supplementary, but I am a bit worried about such a paper writing. While it is good for me to read the manuscript, but it needs to be more compact to make more margins and locate the ablation studies in the manuscript. I personally guess that the authors should have spent lots of efforts in paper writing. The current writing is not good enough for the submission. It needs reformulation.
>
> ---
> ###
> **Response**
>
> We agree that ablation studies are important for validating each component of our approach and for completing the
> reader’s understanding of it. Due to ICLR’s 9-page submission limit, we could not include them earlier. Since the
> rebuttal and camera-ready versions allow up to 10 pages (stated in author-guidelines), we have added an additional page
> and moved the ablation studies into the main paper. Please refer to the updated version.

---

> ### Author Response · Authors · 2025-11-20
> **Response to Reviewer LE1D (w2: Novelty)**
>
> ---
> > **Comment** Even though the authors present the background knowledge for the concept of the embedding alignments in Lines 216-247, it is not something new. More technical speaking, it is simply align the two embedding spaces, one for the embedding vectors from backbone network and the other for the objects' poses from the header network, from the target domain and the source domain. Even, the authors simply leverage the InfoNCE loss. Can the authors provide more clues or claims why the proposed schemes are novel? Moreover, it is not also something new to introduce pseudo labeling in the unsupervised domain adaptation problem. There are lots of related works who also use the same idea, but based on the submission, I cannot clearly tell the differences.
>
> ---
> ###
> **Response**
>
> Each component of CAPLR builds on but goes beyond existing ideas in domain adaptation and contrastive learning by tailoring them specifically to the challenges of 6DoF pose estimation.
>
> ### 1. Feature Alignment via Contrastive Learning
> While contrastive feature alignment is common in tasks such as classification and segmentation, existing formulations cannot be directly applied to 6DoF pose estimation for several reasons:
>
> - **Defining positives and negatives is non-trivial.** In 6DoF pose estimation, the target space is continuous, making the notion of “similar” or “dissimilar” samples far less straightforward than in discrete-label tasks. A naive approach would be to train a model on the synthetic domain, use its predictions on real images, and treat cross-domain samples with similar predicted poses as positives, and others as negatives. However, due to the significant domain gap, these predictions are not reliable, making such pair formation unstable.
>
> - **Motivation for our filtering step.** To address the above issue, we introduce a feature-similarity–based filtering step that stabilizes positive/negative construction across domains. To the best of our knowledge, this strategy has not appeared in prior work, and it is essential for making contrastive alignment feasible.
>
> - **Only task-relevant regions should be aligned.** Unlike classification or segmentation, where the entire feature map contributes to the prediction, 6DoF pose estimation relies only on specific spatial regions corresponding to object projections. Aligning the full feature map introduces noise from background or irrelevant areas and yields limited improvements. To address this, we extract only the task-relevant regions based on the projection of the model’s predicted keypoints. To account for potential localization errors caused by the domain gap, we extract DxD patches rather than single points.
>
> ### 2. Pseudo-Labeling as a Refinement Step
> Pseudo-labeling is widely used in domain adaptation, but our use of it differs in purpose and design.
>
> After contrastive alignment, we noticed that the model still shows prediction uncertainty when the same real image is viewed under small augmentations. In some augmentations, the predicted keypoints are more accurate than in the original view. We exploit this observation by generating pseudo-labels from the model’s predictions across augmented versions of the same image. However, pseudo-labels may be unreliable if the model fails under certain augmentations. To mitigate this, we introduce two filtering mechanisms:
>
> - Confidence filtering, which ensures the model only learns from predictions it is confident about.
> - Spatial consistency filtering, which ensures agreement among augmented predictions before accepting an update.
>
> Together, these filters allow pseudo-labels to function as a refinement step, not an alignment step, which is a key difference from standard pseudo-labeling pipelines.
>
> ### 3. Overall Framework Novelty
> The combination of:
>
> - stable positive/negative construction via feature similarity,
> - task-focused contrastive alignment through region extraction, and
> - augmentation-consistent pseudo-label refinement
>
> forms a coherent and novel framework tailored for 6DoF pose estimation. This design leads to consistent improvements across widely used benchmarks that capture diverse real-world challenges.

---

> ### Author Response · Authors · 2025-11-20
> **Response to Reviewer LE1D (Q: Writing concerns)**
>
> ----
> > **Question**   Overall, it is easy to read the paper, but it needs to be more concise and compact. Ablation studies are not written in the manuscript, but the supplementary material. I am not sure why the Section 3.1 is necessary. The paper needs lots of revision. Beside from the writing, I do not think that the authors present the novel ideas or contributions. There are lots of existing strategy to solve the unsupervised domain adaptation problem for the 6-DoF pose estimation problem. However, I cannot clearly tell the differences between the previous works and the proposed schemes.
>
> ----
> ###
>
> **Response**
>
> In the updated version of the paper, we revised the abstract and introduction based on Reviewer **Givv**’s feedback to better
> highlight the key contributions over the limitations of prior work.
>
> Regarding Section 3.1 (Preliminaries), following prior work, we consider this section important because it clarifies the
> problem setup and establishes the foundation for the methodology.
>
> That said, if any part of the manuscript could be made clearer, we are open to further suggestions on how to improve
> the the clarity and quality of the remaining sections.

---

> > ### Comment · Reviewer_LE1D · 2025-11-26
> > **Additional question to the authors.**
> >
> > Thank you for your precise comments.
> >
> > By the way, I found that I cannot find the updated part in the manuscript. Commonly in the ICLR submission, the authors tend to colorize the font to show which sentences or words are revised during the rebuttal. Once it goes to the camera ready stage, then the authors de-colorize the revised font.
> >
> > This is not a mandatory option, but I would say that it is highly recommended.

---

> ### Comment · Reviewer_LE1D · 2025-11-26
> **Following questions to the authors about the novelty.**
>
> Overall, the authors' rebuttal has __not__ sufficiently addressed my concerns regarding novelty.
>
> First, regarding the difficulty of defining positives/negatives: I do not agree with the authors' assessment, stated as:
>
> > _"the target space is continuous, making the notion of “similar” or “dissimilar” samples far less straightforward than in discrete-label tasks."_
>
> Contrastive loss fundamentally learns relative representations (managing 'close' vs. 'far' relationships) rather than absolute values (i.e., 6DoF). So, the loss is originally computed on top of the continuous space. For example, Open-vocabulary 3D semantic segmentation methods [1, 2, 3] already handle continuous CLIP embeddings effectively. Specifically, [3] utilizes contrastive learning for feature alignment. How does your formulation differ from these existing continuous alignment techniques compared with the proposed submission?"
>
> Second, the motivation that
>
> > _"only task-relevant regions should be aligned"_
>
> is already well-explored. For example, [1] uses SAM and CLIP to extract object-centric masks for region-specific alignment.
>
> Finally, I am concerned about the claim that existing formulations for classification/segmentation cannot apply to 6DoF pose estimation. What specific technical barriers prevent previous algorithms from working in the 6DoF setup? The components presented appear to be general-purpose techniques used in other tasks. __I fail to see how the proposed algorithm is specifically designed to solve unique challenges in 6DoF pose estimation.__
>
> If this theoretical gap is clearly resolved, I would be willing to raise my score. Otherwise, I stand by my opinion regarding the lack of novelty.
>
> References
> [1] LangSplat: 3D Language Gaussian Splatting, CVPR 2024
> [2] RegionPLC: Regional Point-Language Contrastive Learning for Open-World 3D Scene Understanding, CVPR 2024
> [3] Mosaic3D: Foundation Dataset and Model for Open-Vocabulary 3D Segmentation, CVPR 2025

---

> > ### Author Response · Authors · 2025-12-02
> > **Response to Reviewer LE1D (Clarifications on Novelty) (3/5)**
> >
> > > **Comment** Finally, I am concerned about the claim that existing formulations for classification/segmentation cannot apply to 6DoF pose estimation. What specific technical barriers prevent previous algorithms from working in the 6DoF setup?
> >
> > **Response**
> >
> > 1- Segmentation and classification approaches rely on semantic correspondences, which are tolerant to pixel-level inaccuracies [1] (i.e, small spatial errors do not significantly affect the final prediction) . In contrast, 6DoF pose estimation requires geometric correspondences with sub-pixel precision, because even slight spatial errors propagate into large pose errors. This constitutes the key technical barrier: techniques designed for semantic similarity are not strict enough to produce the precise similar/dissimilar pairs needed for geometric feature alignment.
> >
> > 2- The main barrier with SAM-based masking (e.g., in LangSplat[3]) is that SAM highlights the entire visible object and ignores regions that are occluded or affected by strong lighting (Sunlamp domain in SPEED+). As a result, important parts of the object that influence the final pose are missing. Our method highlights only the regions that are actually relevant for the pose task, and it selects them even when they are not visible in the real image. These regions are then pulled toward their correct locations in the synthetic domain, ensuring that both visible and invisible parts are used. This leads to more complete and pose-relevant region selection than SAM.
> >
> > 3-  Methods like FixMatch[4] are built for classification, where labels stay the same under augmentation. In keypoint regression for pose estimation, the targets are continuous and change with image transformations, so standard pseudo-labeling cannot be used directly. Our refinement must therefore enforce geometric validity, which we do by applying two checks: a confidence threshold and a spatial consistency test. These criteria are necessary because keypoint-based pose tasks have very low tolerance for geometric errors.
> >
> > 4- Another barrier is that the reviewer’s suggested methods rely on large foundation-model pipelines such as SAM and CLIP, which require text–image inputs and heavy computation. This is unrealistic for real-time applications such as space-navigation scenarios, where only camera images of the target satellite are available and the system must adapt quickly without labels. Adding text inputs or depending on large models would make the pipeline too slow and impractical.
> >
> >
> >
> > **References**
> >
> > [1] Fan, Zhenfeng, et al. "Dense semantic and topological correspondence of 3D faces without landmarks." Proceedings of the European Conference on Computer Vision (ECCV). 2018.
> >
> >
> > [2] SINGH, Inder Pal, et al. "Bridging the Synthetic-Real Gap: Supervised Domain Adaptation for Robust Spacecraft 6-DoF Pose Estimation." Advanced Space Technologies in Robotics and Automation (2025).
> >
> > [3] LangSplat: 3D Language Gaussian Splatting, CVPR 2024
> >
> > [4] Sohn, Kihyuk, et al. "Fixmatch: Simplifying semi-supervised learning with consistency and confidence." Advances in neural information processing systems 33 (2020): 596-608.

---

> > ### Author Response · Authors · 2025-12-02
> > **Response to Reviewer LE1D (Clarifications on Novelty) (4/5)**
> >
> > > **Comment** The components presented appear to be general-purpose techniques used in other tasks. I fail to see how the proposed algorithm is specifically designed to solve unique challenges in 6DoF pose estimation.
> >
> > **Response**
> >
> > **What is general-purpose**
> >
> > - InfoNCE contrastive loss (e.g., SimCLR)
> > - Pseudo-labeling (common in semi-supervised learning)
> > - Consistency regularization (e.g., FixMatch)
> >
> > **What is specific to our method**
> >
> > - Geometric correspondence across domains: Feature-similarity–based pairing followed by pose-based refinement to produce geometric correspondences between synthetic and real domains which is not addressed in standard contrastive pipelines.
> >
> > - Keypoint-based patch extraction (Fig. 2-B): Instead of using SAM or class-activation maps, we project predicted keypoints onto feature maps and extract localized D×D patches around them.
> >
> > - Contrastive loss is applied to the selected feature patches at two levels of the network (input- and output-proximal), acting as proxies for geometric alignment.
> >
> > - Pseudo-labels are considered only when two task-specific criteria are satisfied: Confidence threshold and Spatial consistency check which are required for 6DoF keypoint regression, where small geometric errors cannot be tolerated.
> >
> > - The combination of the three stages (CDP, LPCA, CBPR) forms a unified pipeline that achieves state-of-the-art performance under comparable UDA settings.

---

> ### Author Response · Authors · 2025-12-01
> **Response to Reviewer LE1D (Q: Highlighting Revisions in the Updated Manuscript)**
>
> > **Comment:**  By the way, I found that I cannot find the updated part in the manuscript. Commonly in the ICLR submission, the authors tend to colorize the font to show which sentences or words are revised during the rebuttal. Once it goes to the camera ready stage, then the authors de-colorize the revised font.
>
> **Response**
>
> Following the reviewer’s recommendation, we have uploaded a new revised version of the paper. All modifications made during the rebuttal period are now clearly highlighted in **blue** throughout the manuscript. This should make it easier to locate all updates and revisions.

---

> ### Author Response · Authors · 2025-12-01
> **Response to Reviewer LE1D (Clarifications on Novelty) (1/5)**
>
> > **Comment** First, regarding the difficulty of defining positives/negatives: I do not agree with the authors' assessment, stated as: "the target space is continuous, making the notion of “similar” or “dissimilar” samples far less straightforward than in discrete-label tasks."
> Contrastive loss fundamentally learns relative representations (managing 'close' vs. 'far' relationships) rather than absolute values (i.e., 6DoF). So, the loss is originally computed on top of the continuous space. For example, Open-vocabulary 3D semantic segmentation methods [1, 2, 3] already handle continuous CLIP embeddings effectively. Specifically, [3] utilizes contrastive learning for feature alignment. How does your formulation differ from these existing continuous alignment techniques compared with the proposed submission?"
>
> **Response**
>
> Contrastive learning operates naturally on continuous feature spaces. Our intent in Sec. 2.1 was not to claim that continuity itself prevents contrastive learning, but that for regression tasks such as 6DoF pose, positive/negative pairs must be defined from **continuous pose similarity**, which is non-trivial  [1,2],  particularly under domain shift and symmetries.
>
>
> In semantic contrastive learning methods (e.g Mosaic3D [3])  pairs are defined based on discrete semantic categories. Whether supervision comes  from labels or from a pretrained language–vision manifold (e.g., CLIP), semantic identity provides a clear criterion for example all regions corresponding to the same concept (text-image, image-image) form positives regardless of their
> appearance, orientation, or shape variation, and this relationship is globally consistent and transitive. The semantic
> embedding space itself provides a reliable supervisory signal that makes pair definition straightforward.
>
> In contrast, 6DoF pose estimation offers no such fixed supervisory manifold. Pair construction must rely on geometric
> similarity in pose space (i.e., closeness in rotation and translation) which is a continuous and task-specific quantity that is
> not directly observable in the unlabeled target domain. Unlike semantic similarity, geometric similarity is not guaranteed
> to correlate with appearance similarity across domains due to texture, lighting, and rendering gaps. Consequently:
>
> &nbsp;&nbsp;&nbsp;&nbsp;&nbsp;&nbsp;1-Visually similar target images may correspond to very different poses.
>
> &nbsp;&nbsp;&nbsp;&nbsp;&nbsp;&nbsp;2-Images with nearly identical poses may look very different across domains.
>
> Therefore, the usual semantic contrastive assumption “visually/semantically similar ⇒ should be close in feature space” becomes unreliable for mining pose-positive pairs in 6DoF pose across domains.
>
> Our method addresses this unique difficulty by introducing a two-stage cross-domain pairing module that first matches
> images via intermediate relevant feature regions and then refines pairs using pose-consistency. This is fundamentally different from the contrastive settings in semantic 3D segmentation, where positive pairs are obtained from semantic labels or CLIP embeddings.
>
> In summary, the novelty of our formulation lies in enabling contrastive alignment without access to semantic labels
> or pretrained pose embeddings, requiring us to learn what constitutes a positive geometric pair under domain shift.
>
> **References**
>
> [1] Zha, Kaiwen, et al. "Rank-n-contrast: learning continuous representations for regression." NeurIPS 2023.
>
> [2] Li, Zhujun, Shuo Zhang, and Ioannis Stamos. "Learning Point Cloud Representations with Pose Continuity for Depth-Based Category-Level 6D Object Pose Estimation." ICCV. 2025.
>
> [3] Mosaic3D: Foundation Dataset and Model for Open-Vocabulary 3D Segmentation, CVPR 2025

---

> ### Author Response · Authors · 2025-12-01
> **Response to Reviewer LE1D (Clarifications on Novelty) (2/5)**
>
> > **Comment** the motivation that "only task-relevant regions should be aligned" is already well-explored. For example, [1] uses SAM and CLIP to extract object-centric masks for region-specific alignment.
>
> **Response**
>
> We do not claim novelty for the general idea of region-of-interest alignment. However (lines 74-80), in most UDA applications for segmentation or classification, it is common to align all features since every feature contributes meaningfully to the output. In contrast, for 6DoF keypoint-based pose estimation, this full-feature alignment fails because the feature maps contain large amounts of irrelevant or background information that do not contribute to the final output. Unlike segmentation, where aligning the entire object region is appropriate and beneficial, 6DoF pose estimation depends on a sparse set of keypoints, and aligning only the regions that directly support these keypoints is crucial for effective domain adaptation. We hypothesize that this fundamental difference is why direct feature alignment has been largely overlooked in previous 6DoF work and is the key motivation behind our region alignment approach.
>
> Regarding novelty, prior works (In segmentation) commonly extract regions of interest **at the image level** using an explicit detector or pretrained segmentation network like SAM in LangSplat[1]. This approach, however, has two important limitations when adapted to UDA for 6DoF pose estimation: (i) it introduces an auxiliary network whose accuracy degrades under domain shift, adding a new, compounding source of error; and (ii) alignment is applied to all pixels in the extracted mask, many of which are irrelevant for sparse keypoint regression.
>
> In contrast, our method extracts task-relevant regions directly from the pose network’s predicted keypoints **in feature
> space**, focusing only on local D×D neighborhoods that influence keypoint heatmaps and pose estimation. This
> detector-free, keypoint-conditioned alignment strategy is both lightweight and tightly coupled to the 6DoF regression
> task, overcoming shortcomings of full-feature or image-level mask alignment which tend to fail under domain shift.
>
> **References**
>
> [1] LangSplat: 3D Language Gaussian Splatting, CVPR 2024

---

> ### Author Response · Authors · 2025-12-02
> **Response to Reviewer LE1D (Clarifications on Novelty) (5/5)**
>
> Beyond conceptual differences, our results demonstrate that existing UDA approaches for 6DoF pose are insufficient:
>
> - LineMOD dataset (Table1): CAPLR 97.2% ADD-S (beats ONDA-Pose 93.2% (CVPR25), Tex-Pose 91.7% (CVPR23),
> MAST 93.2% (IJCAI23). (**+4%** over closest baseline).
>
> -  Occluded-LineMOD (Table2): CAPLR: 85.8% ADD-S (beats ONDA-Pose 69.3%, Tex-Pose 66.7%, MAST 61.4%).
> (**+16.5%** over closest baseline).
>
> - SPEED (Table 3): CAPLR achieves an Epose of 0.101, outperforming larger baselines, and approaching typical real-mission accuracy targets (translation error + σ < 1% of relative distance, rotation error + σ < 3°).

---

### Official Review · Reviewer_rXJD · 2025-11-01

**Soundness:** 3
**Presentation:** 3
**Contribution:** 3
**Rating:** 6
**Confidence:** 3

**Summary:**

This paper aims to solve unsupervised domain adaptation (UDA) for 6-DoF object pose estimation in a sim-to-real setting. The proposed framework, called CAPLR, introduces three main components: 1) Cross-Domain Pairing (CDP) finds source–target pairs that are likely to depict similar object poses, using a two-stage strategy (first a feature/heatmap-based similarity, then a pose-distance filtering) to reduce noisy matches, 2) Local Patch Contrastive Alignment performs contrastive learning both at the backbone level and at the head level so that features useful for pose prediction are actually aligned, not just globally domain-invariant ones, and 3) Consistency-Based Pseudo-Label Refinement (CBPR) generates pseudo labels for target images and refine them via augmentation consistency so that only stable keypoints are kept. The method is evaluated on standard synthetic-to-real 6D benchmarks (e.g., LineMOD, Occluded-LineMOD, HomeBrewedDB, SPEED-like settings) and claims to outperform or match prior UDA approaches under comparable settings.

**Strengths:**

1. Dual-level contrastive learning seems effective.
*  Unlike the previous contrastive learning methods that applied at the backbone level feature distribution, the paper exploits both the backbone and the head level feature distribution for 6D pose estimation.

2. Broad experiment and outperformed performance.
* The proposed method is evaluated on standard synthetic-to-real 6D benchmarks (e.g., LineMOD, Occluded-LineMOD, HomeBrewedDB, SPEED-like settings) and claims to outperform or match prior UDA approaches under comparable settings.

**Weaknesses:**

1. Failure-case analysis of pairing.
* In the ablation where you compare GT pairing vs. your 2-stage pairing, there is a non-trivial gap. Could you show some qualitative or per-scenario analysis, such as under what conditions (occlusion, illumination, near-symmetric poses, scale mismatch) does your pairing fail most often?

2. Extension to category-level 6D.
* The method is demonstrated mainly on instance-level objects (LineMOD family). However, related work cited includes category-level UDA for 6D. If the method is meant to generalize to slightly different CADs/scales (which is a key challenge in category-level 6D), there should be experiments or at least analysis on how pairing behaves in that setting. If not, can you discuss how the framework can be extended to category-level 6D pose estimation?

**Questions:**

Please answer the weakness parts.

---

> ### Author Response · Authors · 2025-11-20
> **Response to Reviewer rxJD (w1: Failure-case analysis of pairing)**
>
> We thank the reviewer for the constructive feedback and suggestions. We address the concerns below and have uploaded a revised version of the paper reflecting all reviewers  comments.
>
> ---
> > **Comment** In the ablation where you compare GT pairing vs. your 2-stage pairing, there is a non-trivial gap. Could you show some qualitative or per-scenario analysis, such as under what conditions (occlusion, illumination, near-symmetric poses, scale mismatch) does your pairing fail most often?
> ---
>
> ###
>
> **Response**
>
> To address the concerns regarding the pairing strategy, we added a detailed analysis in the supplementary material (Supplementary Section 6.3.3 pp. 25–26) of the revised paper, where our cross-domain pairings are directly compared with those obtained using ground-truth labels. We invite the reviewer to consult these pages for a thorough illustration and discussion.
>
> First, we analyze the rotational and translational differences between paired samples for both our method and the ground-truth pairings, visualizing them together in the same scatter plots (Figure 12). As you noted in your review, the evaluation results in Table 5 alone do not clearly expose the relative quality of the two pairing strategies. However, the new plots show that while our CDP-based pairing is not perfect and introduces some noisy pairs, these constitute only a small portion of the total. Using ground-truth labels, most valid pairs fall within the region of rotation difference $\leq$ 25° and translation difference $\leq$ 1 m. Our CDP stage achieves 90.8\% of pairs within this range on Lightbox and 81.9\% on Sunlamp, demonstrating that the learned pairing strategy is effective and closely aligned with the ground-truth pairing distribution.
>
>
> We also expanded our analysis with a study of failure cases. We selected five of the worst CDP-generated pairs and examined the real–synthetic correspondences. These failures predominantly occur in situations where the object is heavily affected by shadows, harsh lighting, or partial occlusions conditions that suppress distinctive features and make pairing inherently difficult. Interestingly, even for these poorly paired examples, our full framework still improves pose predictions. This improvement is largely attributed to the knowledge transferred from correctly aligned pairs, which helps the model generalize despite occasional pairing errors.
>
> Overall, the new analysis confirms that although CDP may produce a few noisy pairs, the vast majority are accurate and closely reflect the ground-truth pairing structure, and the framework remains robust even in failure cases.

---

> > ### Comment · Reviewer_rXJD · 2025-11-26
> >
> > Thanks for the responses. The responses resolved well my concerns. I will keep my initial decision.

---

> ### Author Response · Authors · 2025-11-20
> **Response to Reviewer rxJD (w2: Extension to category-level 6D)**
>
> ---
> > **Comment** The method is demonstrated mainly on instance-level objects (LineMOD family). However, related work cited includes category-level UDA for 6D. If the method is meant to generalize to slightly different CADs/scales (which is a key challenge in category-level 6D), there should be experiments or at least analysis on how pairing behaves in that setting. If not, can you discuss how the framework can be extended to category-level 6D pose estimation?
> ---
> ###
>
> **Response**
>
> Our current experiments focus on instance-level objects, specifically using the LineMOD dataset family. This choice allows us to clearly evaluate the effectiveness of our Unsupervised Domain Adaptation (UDA) framework, which combines contrastive alignment and pseudo-label refinement. We follow the same controlled UDA settings as prior studies to ensure a fair comparison and to better understand domain shifts and feature alignment.
>
> We understand that category-level 6D pose estimation is more challenging because objects within a category can look different in shape and size, and real images may contain unseen objects. Our current work does not directly address these challenges.
>
> The pairing stage, which matches images based on keypoint predictions, would need improvements to handle diverse CAD models within a category, because shapes can vary significantly and make direct keypoint matching ambiguous. To handle this, future work could use category-level representations, like canonical keypoints or learned embeddings that capture shape differences more generally.
>
> We see extending our framework to category-level 6D pose estimation as an important direction for future research.
>
>
>
> Finally, we thank the reviewer for their feedback and remain open to any further suggestions or questions.

---

### Official Review · Reviewer_Givv · 2025-11-03

**Soundness:** 3
**Presentation:** 3
**Contribution:** 2
**Rating:** 6
**Confidence:** 4

**Summary:**

The paper addresses unsupervised domain adaptation for 6D object pose estimation and introduces CAPLR, which consists of three key components, including cross-domain pairing, contrastive alignment, and consistency-based pseudo-label refinement. Experiments on LineMOD, LineMOD-Occlusion, and SPEED+ demonstrate the effectiveness of the proposed approach.

**Strengths:**

- Instead of relying solely on self-supervised learning, CAPLR employs contrastive alignment and pseudo-label refinement for UDA pose estimation, with ablation studies validating the effectiveness of each component.
- CAPLR realizes SOTA on UDA setting on LineMOD, LineMOD-Occlusion, and SPEED+ .

**Weaknesses:**

- For existing methods, the paper does not sufficiently analyze their limitations in the abstract and introduction. In particular, the discussion of self-supervised learning is too general, using broad statements such as “Many approaches rely solely on self-supervision in the target domain, which has inherent limitations” and “However, these methods are limited when the domain gap is large,” without providing concrete analysis. Consequently, the advantages of the proposed method over these limitations are not clearly highlighted in these sections.
- For cross-domain pairing, I have the following concerns:
    - I am concerned about the robustness of the confidence-weighted embeddings. In keypoint regression, the features of different keypoints are expected to remain distinguishable; however, this design relies on the consistency of both keypoint features and confidence scores across domains. The results in Table 5 do not provide ablation studies to assess this, for example, comparisons with simple average pooling over keypoint features or using masked foreground features. Considering the widespread use of DINOv2 features for template matching in pose estimation, would it be more robust to instead use the CLS tokens from DINOv2?
    - In Table 5, the results based solely on pose prediction outperform those using feature-based selection. Why not reverse the order of the two steps, i.e., perform pose-prediction-based selection first, followed by feature-based selection?
    - Each target image is paired with an optimal source image. Is there a filtering strategy to avoid highly mismatched pairs?
- For pseudo label refinement,  I have the following concerns:
    - What are the augmentations used?
    - Since $L_{task}$ in Eq. (10) is not used at this stage, i.e., no ground-truth supervision is applied, how is it ensured that training progresses in the correct direction rather than falling into shortcuts? Could you provide the performance on the test or validation sets across training epochs?
- For the results in Tables 1 and 2, it is recommended to provide the lower and upper bound performance of CAPLR, i.e., using only S and using S+R, to better illustrate the gains attributable to the UDA method rather than the baseline model.

**Questions:**

- For the source domain, why are the predicted poses and keypoints used in cross-domain pairing and contrastive alignment instead of the ground-truth annotations?

---

> ### Author Response · Authors · 2025-11-20
> **Response to Reviewer Givv (w1: Writing Improvements)**
>
> We thank the reviewer for the constructive feedback and suggestions. We address the concerns below and have uploaded a revised version of the paper reflecting all reviewers comments.
>
> ---
> > **Comment** For existing methods, the paper does not sufficiently analyze their limitations in the abstract and introduction. In particular, the discussion of self-supervised learning is too general, using broad statements such as “Many approaches rely solely on self-supervision in the target domain, which has inherent limitations” and “However, these methods are limited when the domain gap is large,” without providing concrete analysis. Consequently, the advantages of the proposed method over these limitations are not clearly highlighted in these sections.
> ---
>
> ###
> **Response**
>
> Based on the reviewer’s feedback, we have updated both the Abstract and Introduction in the revised paper. The advantages of our method over the limitations of prior works are now communicated more clearly.

---

> ### Author Response · Authors · 2025-11-20
> **Response to Reviewer Givv (w2: Cross-domain pairing concerns 1/4)**
>
> ---
> > **Comment** I am concerned about the robustness of the confidence-weighted embeddings. In keypoint regression, the features of different keypoints are expected to remain distinguishable; however, this design relies on the consistency of both keypoint features and confidence scores across domains. The results in Table 5 do not provide ablation studies to assess this, for example, comparisons with simple average pooling over keypoint features or using masked foreground features.
> ---
>
> ###
> **Response**
>
> Our method uses keypoint confidence scores to adaptively weigh keypoint-related areas from the feature map. This allows regions linked to high-confidence keypoints to contribute more, while regions associated with unreliable or occluded keypoints will have less influence in the candidate selection phase. This design assumes reasonable consistency in both region quality and confidence estimation in both the source and target domains. We acknowledge that this assumption may not hold when confidence scores are inaccurate, and such cases require further analysis.
>
> As suggested by the reviewer, we further evaluated Cross-domain pairing performance by using average pooling, i.e.,
> treating all areas equally without any confidence weighting. The overall performance (Table below) remains comparable; however,
> confidence weighting provides a clearer advantage in identifying correct matches under rotation, an aspect more critical
> to our setup than translation. While patch based keypoint projection can handle small positional shifts between the
> pairs, maintaining rotation consistency is essential for accurately matching corresponding areas.
>
>
> **Table**
> **Table: CDP Performance**
>
> | **Method**              | **Lightbox** [0.5m,10°] ↑ | **Lightbox** [1m,10°] ↑ | **Lightbox** [0.5m,15°] ↑ | **Lightbox** [1m,15°] ↑ | **Sunlamp** [0.5m,10°] ↑ | **Sunlamp** [1m,10°] ↑ | **Sunlamp** [0.5m,15°] ↑ | **Sunlamp** [1m,15°] ↑ |
> |-------------------------|---------------------------|---------------------------|----------------------------|---------------------------|---------------------------|---------------------------|----------------------------|---------------------------|
> | **confidence weighted (ours)** | 37.83                 | **56.88**                 | 59.63                      | **83.26**                 | 32.78                 | **46.76**                 | 52.38                      | **72.62**                 |
> | **avg-pooling**         | **41.69**                     | 49.45                     | **72.02**                      | 82.12                     | **34.22**                     | 38.98                     | **62.92**                      | 71.16                     |

---

> ### Author Response · Authors · 2025-11-20
> **Response to Reviewer Givv (w2: Cross-domain pairing concerns 2/4)**
>
> ---
> > **Comment** Considering the widespread use of DINOv2 features for template matching in pose estimation, would it be more robust to instead use the CLS tokens from DINOv2?
> ---
>
> ###
> **Response**
>
> Using DINOv2 CLS tokens for template matching is an interesting alternative that benefits from the global image context and has shown strong performance in various vision tasks. However, recent work [1] demonstrates that particularly in large DINOv2 models, there can be a disconnect between what CLS token encodes and the localized patch features. Specifically, this paper warns that the CLS token may not faithfully reflect local, spatially detailed content. Which means that although the CLS token provides a good global descriptor, it may not retain the detailed spatial information necessary for precise tasks like 6D pose estimation.
>
> Our proposed method instead, focuses on leveraging spatially localized keypoint features that directly correspond to object parts critical for precise pose regression. Integrating or comparing with CLS token-based representations is an exciting direction for future work to enhance robustness further.
>
>
> **References**
>
> [1] Lappe, Alexander, and Martin A. Giese. "Register and CLS tokens yield a decoupling of local and global features in large ViTs." arXiv preprint arXiv:2505.05892 (2025).

---

> ### Author Response · Authors · 2025-11-20
> **Response to Reviewer Givv (w2: Cross-domain pairing concerns 3/4)**
>
> ---
> > **Comment** In Table 5, the results based solely on pose prediction outperform those using feature-based selection. Why not reverse the order of the two steps, i.e., perform pose-prediction-based selection first, followed by feature-based selection?
> ---
>
> ###
> **Response**
>
> Our pairing procedure is performed in two stages: first, feature-based filtering to reduce candidate pairs, followed by final selection based on predicted pose.
>
> The reason behind the above order is that the domain gap can lead the model to incorrectly predict poses, causing mismatches with unrelated synthetic ones. To overcome this, we initially narrow down the candidates using deep features, which provide geometric and semantic data to exclude dissimilar options while keeping likely matches. Later we consider images with similar poses, making pose similarity crucial for selection. Table 5 confirms the effectiveness of this strategy.
>
> Reversing the sequence to first filter by predicted pose, then by features, is ineffective. Features are too abstract to be a reliable final criterion. As shown in the table below, using pose to narrow down candidates followed by feature-based selection results in noisy outcomes, failing to match the quality of pairs achieved with our current method.
>
>
>
>
> **Table**
> **Table: CDP Performance**
>
> | **Method**        | **Lightbox** [0.5m,10°] ↑ | **Lightbox** [1m,10°] ↑ | **Lightbox** [0.5m,15°] ↑ | **Lightbox** [1m,15°] ↑ | **Sunlamp** [0.5m,10°] ↑ | **Sunlamp** [1m,10°] ↑ | **Sunlamp** [0.5m,15°] ↑ | **Sunlamp** [1m,15°] ↑ |
> |-------------------|---------------------------|---------------------------|----------------------------|---------------------------|---------------------------|---------------------------|----------------------------|---------------------------|
> | **Two-stage (ours)**     | 37.83                 | 56.88                 | 59.63                      | 83.26                 | 32.78                 | 46.76                 | 52.38                      | 72.62               |
> | **Inverted Two-stage** | 7.80                      | 17.51                     | 16.13                      | 36.16                     | 4.98                      | 12.47                     | 11.90                      | 28.20                     |

---

> ### Author Response · Authors · 2025-11-20
> **Response to Reviewer Givv (w2: Cross-domain pairing concerns 4/4)**
>
> ---
> > **Comment** Each target image is paired with an optimal source image. Is there a filtering strategy to avoid highly mismatched pairs?
> ---
>
> ###
> **Response**
>
> In the current CDP stage, there is no explicit mechanism to filter or remove highly mismatched pairs; all discovered pairs are used during adaptation. However, the number of mismatched pairs is minimal.
>
> To quantify this, we added a new analysis in the supplementary material (Section 6.3.3, pages 25–26) comparing the discovered pairs with the ground truth. In the Lightbox domain, over 90\% of pairs fall within the acceptable range of ground-truth differences, and in the Sunlamp domain, over 80\% do. we did this experiments mainly on the SPEED+ dataset because the SPEED+ dataset has more random pose labels (with different lighting and background conditions) compated to LM/LM-O datasets which are actually a sequence of images from a trajectory.
>
> In addition, we include a failure-case analysis in the same section. Even when pairs are highly mismatched, the model typically still improves the pose after adaptation. This is likely due to the model’s generalization and the knowledge learned from other well-aligned pairs.
>
> We invite the reviewer to refer to the new added Section 6.3.3 for detailed information and clarifications.

---

> ### Author Response · Authors · 2025-11-20
> **Response to Reviewer Givv (w3: Pseudo label refinement concerns 1/2)**
>
> ---
> > **Comment** What are the augmentations used?
> ---
>
> ###
> **Response**
>
> We thank the reviewer for pointing this out. In the original submission, we missed including  details of the augmentations used during the CBPR stage. We have now clarified this in the revised manuscript (lines 909-910).
>
> For CBPR, we intentionally employ only mild augmentations that do not alter the geometric structure or discriminative features of the object, ensuring that the model’s predictions remain valid. Specifically, we use blur and noise–based augmentations from the Albumentations library (e.g., Blur, Emboss, FancyPCA). These augmentations introduce slight appearance variations while preserving the underlying pose and keypoint information, which is crucial for stable pseudo-label refinement.
>
> Based on the reviewer's feedback, this information has been added to the revised paper for full transparency.

---

> ### Author Response · Authors · 2025-11-20
> **Response to Reviewer Givv (w3: Pseudo label refinement concerns 2/2)**
>
> ---
> > **Comment** Since **L_task** in Eq. (10) is not used at this stage, i.e., no ground-truth supervision is applied, how is it ensured that training progresses in the correct direction rather than falling into shortcuts? Could you provide the performance on the test or validation sets across training epochs?
> ---
>
> ###
> **Response**
>
> Indeed, the refinement step without task supervision could raise concerns regarding stability or potential performance degradation. However, this stage ensures stable learning through two key mechanisms:
>
> - **Familiar and mild augmentations:** We use augmentations that the model has seen during training and simple transformations that do not alter the object’s discriminative features (e.g., blur, noise, flipping). This ensures that the model continues to produce valid predictions, keeping the pseudo-labels within the knowledge learnt in previous steps.
> - **Strict pseudo-label update criteria:** Updates are accepted only if two conditions are satisfied: confidence and spatial consistency. This guarantees that the model retains its prior knowledge while it is refined slightly to reduce uncertainty. As shown in Section 6.1.2 of the supplementary material, the number of accepted keypoints increases over epochs, indicating that the model consistently produces reliable predictions across augmentations.
>
> To further confirm training stability, we added Section 6.2.2 in the supplementary material, reporting the percentage of correct keypoints (PCK) on the SPEED+ validation sets (Lightbox and Sunlamp) during this stage. As shown in Figure 4, the PCK remains stable with minimal changes and very small standard deviation, demonstrating that this refinement step does not degrade performance.
>
> Please note that these minimal changes in PCK do not fully reflect pose accuracy, since only the top confident keypoints are considered and a keypoint is marked correct if it is within 5% distance from the ground truth. For this reason, we use the number of accepted keypoints as the metric to select the best checkpoint.

---

> ### Author Response · Authors · 2025-11-20
> **Response to Reviewer Givv (w4: Upper-lower bounds comparison)**
>
> ---
> > **Comment** For the results in Tables 1 and 2, it is recommended to provide the lower and upper bound performance of CAPLR, i.e., using only S and using S+R, to better illustrate the gains attributable to the UDA method rather than the baseline model.
> ---
>
> ###
> **Response**
>
> Following the reviewer's suggestion, which also aligns with Reviewer **jkYC** comments, we added a comparison between a model trained only on synthetic data and a model trained on synthetic + real data in the revised supplementary (Section 6.1.4 page 16). Below is a brief summary of these results.
>
> The lower bound corresponds to training on synthetic images only. The upper bound corresponds to training on synthetic images followed by fine-tuning on real images for the same number of alignment epochs used in CAPLR. For LineMOD and Occluded-LineMOD, fine-tuning uses the same real images available for adaptation; for SPEED+, we use the first 20\% of the real images.
>
> As shown in the tables below, CAPLR consistently delivers substantial improvements over the synthetic-only baseline. In most cases, the gain from synthetic-only → CAPLR is larger than the gap between CAPLR and the upper-bound (synthetic + real) model. This demonstrates that CAPLR effectively closes much of the performance gap to real data while requiring only minimal real imagery.
>
>
>
> ---
> **Tables:**
>
> **Table: Results on LineMOD – Baseline (S), CAPLR (S+R⁻), and Real-Finetuned Model (S+R)**
>
> | Method      | Data   | Ape  | Bvise | Cam  | Can  | Cat  | Drill | Duck | Eggbox | Glue | Holep | Iron | Lamp | Phone | **Mean** ↑ |
> |------------|--------|------|-------|------|------|------|-------|------|--------|------|-------|------|------|-------|------------|
> | **CAPLR_s**| S      | 49.7 | 94.5  | 74.1 | 93.3 | 95.1 | 94.3  | 78.3 | 96.9   | 84.2 | 91.1  | 98.7 | 63.8 | 82.8  | 84.4       |
> |            | S+R⁻   | 82.1 | 95.2  | 93.2 | 97.4 | 98.8 | 97.0  | 89.5 | 100.0  | 96.9 | 95.1  | 98.8 | 78.0 | 93.7  | 93.5       |
> |            | S+R    | 95.1 | 98.9  | 95.1 | 95.8 | 93.1 | 99.7  | 96.3 | 100    | 96.1 | 98.7  | 99.8 | 86.6 | 96.8  | 96.3       |
> | **CAPLR_m**| S      | 75.2 | 99.5  | 85.1 | 95.6 | 96.6 | 95.1  | 96.0 | 99.8   | 84.6 | 92.9  | 99.7 | 75.2 | 90.9  | 90.8       |
> |            | S+R⁻   | 90.7 | 99.5  | 93.2 | 97.9 | 99.3 | 97.7  | 98.8 | 100.0  | 99.5 | 97.0  | 99.9 | 94.1 | 96.6  | 97.2       |
> |            | S+R    | 98.1 | 99.9  | 97.7 | 98.4 | 100  | 99.9  | 99.9 | 100    | 98.6 | 99.7  | 100  | 96.8 | 99.9  | 99.2       |
>
>
>
> **Table: Results on Occluded-LineMOD – Baseline (S), CAPLR (S+R⁻), and Real-Finetuned Model (S+R)**
>
> | Method      | Data   | Ape  | Can  | Cat  | Drill | Duck | Eggbox | Glue | Holep | **Mean** ↑ |
> |------------|--------|------|------|------|-------|------|--------|------|-------|------------|
> | **CAPLR_s**| S      | 54.6 | 80.3 | 56.1 | 79.6  | 55.8 | 91.1   | 62.9 | 80.3  | 70.3       |
> |            | S+R⁻   | 62.0 | 91.3 | 61.2 | 86.1  | 69.4 | 95.6   | 75.6 | 93.6  | 79.3       |
> |            | S+R    | 85.0 | 95.8 | 87.7 | 90.3  | 82.9 | 95.3   | 80.6 | 98.4  | 89.5       |
> | **CAPLR_m**| S      | 54.6 | 84.0 | 54.5 | 92.5  | 79.5 | 97.4   | 68.6 | 86.7  | 77.2       |
> |            | S+R⁻   | 67.0 | 93.9 | 71.3 | 95.0  | 83.3 | 98.0   | 85.8 | 92.4  | 85.8       |
> |            | S+R    | 90.0 | 97.4 | 95.1 | 96.8  | 88.4 | 98.4   | 87.9 | 99.0  | 94.1       |
>
>
> **Table: SPEED+ – Baseline (S), CAPLR (S+R⁻), and Real-Finetuned Model (S+R)**
>
> | Method      | Data | Lightbox  $E_T / E_T^{norm}$ ↓ | Lightbox  $E_R$ ↓ | Lightbox $E_{pose}$ ↓ | Sunlamp $E_T / E_T^{norm}$ ↓ | Sunlamp $E_R$ ↓ | Sunlamp $E_{pose}$ ↓ | Mean $E_{pose}$ ↓ |
> |------------|------|-------------------------------|-----------------|----------------------|--------------------------------|----------------|---------------------|-----------------|
> | **CAPLR_s**| S    | 0.221 / 0.036                 | 8.98            | 0.193                | 0.426 / 0.065                  | 18.93          | 0.395               | 0.294           |
> |            | S+R⁻ | 0.173 / 0.029                 | 7.80            | 0.165                | 0.215 / 0.035                  | 11.17          | 0.230               | 0.198           |
> |            | S+R  | 0.127 / 0.020                 | 4.74            | 0.103                | 0.131 / 0.021                  | 5.38           | 0.115               | 0.109           |
> | **CAPLR_m**| S    | 0.162 / 0.026                 | 5.31            | 0.119                | 0.288 / 0.044                  | 11.09          | 0.237               | 0.178           |
> |            | S+R⁻ | 0.103 / 0.018                 | 3.54            | 0.080                | 0.126 / 0.022                  | 5.77           | 0.122               | 0.101           |
> |            | S+R  | 0.097 / 0.016                 | 2.99            | 0.068                | 0.106 / 0.017                  | 3.69           | 0.082               | 0.075           |

---

> ### Author Response · Authors · 2025-11-20
> **Response to Reviewer Givv (Q: Rationale for Using synthetic prediction rather than groundtruth)**
>
> ---
> > **Question** For the source domain, why are the predicted poses and keypoints used in cross-domain pairing and contrastive alignment instead of the ground-truth annotations?
> ---
>
> ###
> **Response**
>
> During the CDP and LPCA stages, we use model predictions on the synthetic domain instead of ground-truth labels for the following reasons:
>
> - Models trained on the synthetic domain produce keypoint heatmaps and pose predictions that reflect their internal feature embeddings. These predictions are more aligned with the feature space than raw ground-truth coordinates, helping maintain consistency between feature and output distributions during alignment.
> - The model provides confidence scores for its predictions, reflecting uncertainty due to occlusions or other ambiguities. Ground-truth labels, by contrast, always assume perfect confidence and do not convey uncertainty. Using predicted heatmaps allows the alignment process to account for this uncertainty, making it more robust and avoiding unrealistic exact matches that the model may not be able to achieve.
> - The model’s predictions on synthetic images are already highly accurate, often very close to ground-truth labels. This ensures that using predicted outputs does not compromise alignment quality while retaining the benefits of confidence-aware, robust adaptation.
>
>
>
> Finally, we thank the reviewer for the insightful feedback and questions. We hope our responses have addressed them, and we remain open to any further suggestions or clarifications.

---

### Author Response · Authors · 2025-12-03
**Summary of Revisions and Clarifications**

We thank all reviewers for taking the time to read and evaluate our work. The feedback has been very helpful for making the paper more complete and for better explaining what CAPLR does and how it addresses 6DoF pose estimation under domain shift. We also appreciate the positive remarks on the different components of our framework and the experimental rigor across the range of benchmarks we considered.

Through this rebuttal process, we have improved the conceptual clarity, presentation, and empirical validation of our contribution.

**Summary of Key Revisions**

**1- Addressing Experimental Concerns**

- Added evaluation on the SHIRT dataset to validate out-of-distribution generalization and directly address adaptation–test overlap concerns raised for SPEED+ **[jkYC]**.
- Included explicit upper and lower bound comparisons (S, S+R⁻, S+R) to clearly isolate the UDA gains of CAPLR from the synthetic-only baseline and a supervised synthetic+real  trained model **[Givv & jkYC]**.
- Added experiments on the T-LESS dataset (Sequence 4) to further validate the method on highly symmetric, visually similar objects **[jkYC]**.

**2- Addressing Presentation Concerns**
- Moved key ablation studies from the supplementary material into the main paper (using the additional page allowance) so that the contribution of each component is clearly visible and easier to follow **[LE1D & jkYC]**.
- Revised the Abstract and Introduction to more concretely articulate the specific limitations of prior UDA methods for 6DoF pose estimation and how CAPLR addresses these limitations **[Givv]**.
- Clarified previously missing details for readers, including the augmentations used in the CBPR stage **[Givv]**, the presence of symmetric objects in the LineMOD family **[jkYC]**, and the dataset setups/training protocols for methods compared on SPEED+ **[jkYC]**.

**3- Addressing Technical Clarity**
- Added ablation studies comparing confidence-weighted embeddings with simpler alternatives such as average pooling, to justify the design choices in cross-domain pairing **[Givv]**.
- Clarified the rationale for using model predictions (with confidence) rather than ground-truth annotations in cross-domain pairing and contrastive alignment, emphasizing consistency with the model’s internal feature space and uncertainty **[Givv]**.
- Added a detailed failure-case analysis in the supplementary material, comparing learned cross-domain pairs with ground-truth pairings and examining challenging conditions (e.g., occlusion, harsh lighting) and their effect on the framework performance **[Givv & rXJD]**.
- Included an analysis of training stability during pseudo-label refinement using PCK curves across epochs, showing that CBPR remains stable and progressively increases the number of reliable keypoints **[Givv]**.

**4- Addressing Novelty Concerns**
- Clarified the technical novelty of the three components of CAPLR (CDP, LPCA, CBPR) relative to contrastive and pseudo-labeling methods in other domains/tasks suggested by the reviewers, explaining why these cannot be directly applied to 6DoF pose UDA without modification **[LE1D]**.
- Highlighted how CAPLR specifically tackles geometric, pose-regression–specific challenges: constructing cross-domain geometric correspondences under domain shift and symmetry (CDP), aligning only keypoint-conditioned local patches at backbone and head levels (LPCA), and enforcing task-specific confidence and spatial-consistency criteria for pseudo-label refinement (CBPR)  **[LE1D]**.


----
We have addressed each reviewer’s specific questions and concerns in dedicated response threads with these new experiments, justifications, and analyses. We believe these revisions substantially strengthen the paper’s conceptual contribution, presentation, experimental validation, and technical clarity while preserving the core innovations of CAPLR.


Sincerely,

Authors of CAPLR

---

### Meta-Review · Area_Chair_cuZ8 · 2026-01-06

**Summary:**

While the reviewers acknowledge the effectiveness of the proposed method, they express concerns related to the discussion of existing methods, the justification/evaluation of some of the technical components of the method, the generalization of the method to the category-level scenario, the technical novelty of the proposed approach, and some aspects of the evaluation of the method.

**Reviewer Concerns:**

The authors provided a thorough rebuttal, which addressed most of the concerns of Reviewers Givv and rXJD. However, based on the discussion, Reviewer LE1D remained unconvinced about the novelty of the approach, and it is unclear if Reviewer jkYC was fully satisfied when it comes to their concerns about the evaluation of the method, although the reviewer acknowledged the new results on SHIRT.

**Reviewer Scores:**

Two reviewers recommended borderline acceptance, and two reviewers rejection (borderline in one case). One of the positive reviewers mentioned they would maintain their score, and the most negative reviewer was unconvinced by the authors' feedback. The AC does not expect that Reviewer Givv would have increased their score, and it is unclear if Reviewer jkYC would have turned to recommending acceptance. As such, and considering that the rebuttal involved many changes in the paper, the AC argues that the paper is not ready for acceptance but recommends the author to revise it and resubmit to a future venue.

---

### Decision · Program_Chairs · 2026-01-26

Reject